# Maternally inherited piRNAs direct transient heterochromatin formation at active transposons during early *Drosophila* embryogenesis

**Martin H Fabry, Federica A Falconio, Fadwa Joud, Emily K Lythgoe, Benjamin Czech\*, Gregory J Hannon\***

CRUK Cambridge Institute, University of Cambridge, Li Ka Shing Centre, Cambridge, United Kingdom

**Abstract** The PIWI-interacting RNA (piRNA) pathway controls transposon expression in animal germ cells, thereby ensuring genome stability over generations. In *Drosophila*, piRNAs are intergenerationally inherited through the maternal lineage, and this has demonstrated importance in the specification of piRNA source loci and in silencing of *I*- and *P*-elements in the germ cells of daughters. Maternally inherited Piwi protein enters somatic nuclei in early embryos prior to zygotic genome activation and persists therein for roughly half of the time required to complete embryonic development. To investigate the role of the piRNA pathway in the embryonic soma, we created a conditionally unstable Piwi protein. This enabled maternally deposited Piwi to be cleared from newly laid embryos within 30 min and well ahead of the activation of zygotic transcription. Examination of RNA and protein profiles over time, and correlation with patterns of H3K9me3 deposition, suggests a role for maternally deposited Piwi in attenuating zygotic transposon expression in somatic cells of the developing embryo. In particular, robust deposition of piRNAs targeting *roo*, an element whose expression is mainly restricted to embryonic development, results in the deposition of transient heterochromatic marks at active *roo* insertions. We hypothesize that *roo*, an extremely successful mobile element, may have adopted a lifestyle of expression in the embryonic soma to evade silencing in germ cells.

**\*For correspondence:**
benjamin.czech@cruk.cam.ac.uk (BC);
greg.hannon@cruk.cam.ac.uk (GJH)

**Competing interests:** The authors declare that no competing interests exist.

## Introduction

Transposable elements (TEs) are mobile genomic parasites that can change their genomic position or increase in copy number, and therefore pose a threat to genome integrity. Many TEs have evolved mechanisms that promote their activity specifically in gonads, thereby introducing new insertions that are inherited by future generations (*Kim et al., 1994*; *Leblanc et al., 2000*; *Wang et al., 2018*). Accumulation of insertional mutations in germ cells can lead to decreased population fitness and increased risk of disease (*Hancks and Kazazian, 2016*; *Payer and Burns, 2019*). Germ cells, however, harbor protective systems that substantially decrease the likelihood of transposition events.

In animal gonads, the main transposon defense mechanism is the PIWI-interacting RNA (piRNA) pathway (reviewed in *Czech et al., 2018*; *Ozata et al., 2019*). At its core, this system depends on 23- to 30-nt piRNAs to distinguish transposon-derived RNAs from host-encoded transcripts and to direct their associated PIWI proteins to active TE targets. In *Drosophila*, PIWI-guided repression involves cytoplasmic post-transcriptional mRNA cleavage by Aubergine (Aub) and Argonaute-3 (Ago3) and nuclear P-element-induced wimpy testis (Piwi) that engages nascent transposon transcripts and instructs co-transcriptional gene silencing (coTGS) through heterochromatin formation.

**eLife digest** Maintaining the integrity of DNA, which encodes all of the instructions necessary for life, is essential for ensuring the survival of a species, especially when genetic information is transferred across generations. DNA, however, contains selfish, mobile elements, called transposons, that move around the genome, hence their nickname 'jumping genes'. Their movement, a process by which these elements also multiply within genomes, can muddle an organism's DNA if the transposon happens to land in the middle of a gene, creating a mutation which renders the gene inactive. Transposons have also been linked to the development of cancer, which is a group of diseases driven by accumulating genetic mutations.

Animals have evolved various ways of protecting their DNA against transposons. These are especially important in developing egg cells and sperm, known collectively as germ cells. These cells can produce small fragments of RNA, a molecule similar to DNA, which are able to identify and disarm transposons. While it is known that these small RNAs effectively protect adult gonads from DNA damage, it has been unclear how germ cells formed during the beginning of life are protected.

To find out more, Fabry et al. used a combination of genetic sequencing, protein binding and imaging studies to look at the activity of small RNAs, called piRNAs, which are passed on from the mother to her progeny.

By studying the gene expression levels in fruit fly embryos, Fabry et al. showed that certain transposons become highly active in the first few hours of embryo development, posing a potential threat to DNA integrity. The experiments also identified clear signs in the embryos of an active mechanism for controlling transposons that resembles the small RNA system known from adult germ cells. Fabry et al. removed the piRNAs from the embryos and found that without piRNAs, transposons were more active. This indicates a direct role of these small RNAs in controlling transposons in early development and evidence for a maternally inherited defence system in early embryos.

This study provides insights into the control of transposons in fly embryos. More research is needed to find out whether these embryonic mechanisms are conserved in other animals, including humans. Studying the intrinsic mechanisms that prevent DNA damage and protect our genome could, in time, help to identify new approaches to possibly treat and prevent diseases involving genetic mutations.

coTGS requires additional factors acting downstream of Piwi, including Panoramix (Panx), Nuclear Export Factor 2 (Nxf2), NTF2-related export protein 1 (Nxt1), and Cut-up (Ctp), that together form the PICTS complex (also known as SFiNX) (*Batki et al., 2019*; *Eastwood et al., 2021*; *Fabry et al., 2019*; *Murano et al., 2019*; *Schnabl et al., 2021*; *Sienski et al., 2015*; *Yu et al., 2015*; *Zhao et al., 2019*). PICTS interfaces with general chromatin silencing factors including Su(var)205/HP1a, SETDB1/Eggless (Egg), and its co-factor Windei (Wde), Su(var)3–3/Lsd1, and its co-factor coRest, Mi-2, Rpd3, Ovaries absent, and Su(var)2–10 (*Czech et al., 2013*; *Handler et al., 2013*; *Koch et al., 2009*; *Muerdter et al., 2013*; *Mugat et al., 2020*; *Ninova et al., 2020*; *Osumi et al., 2019*; *Rangan et al., 2011*; *Sienski et al., 2015*; *Yang et al., 2019*; *Yu et al., 2015*). While the detailed mechanisms of transcriptional silencing remain to be established, loci targeted by Piwi are decorated in repressive chromatin marks including trimethylated H3K9 (H3K9me3) (*Klenov et al., 2014*; *Le Thomas et al., 2013*; *Rozhkov et al., 2013*; *Sienski et al., 2012*; *Wang and Elgin, 2011*). Loss of Piwi in *Drosophila* ovaries results in de-repression of TEs and correlates with a severe reduction in H3K9me3 deposition at their corresponding genomic loci.

Piwi and Aub, and to a lesser degree Ago3, have been detected as maternally deposited proteins in *Drosophila* embryos (*Brennecke et al., 2007*; *Brennecke et al., 2008*; *Gunawardane et al., 2007*; *Mani et al., 2014*; *Megosh et al., 2006*; *Rouget et al., 2010*). Considering that pluripotent progenitor cells give rise to multiple cell lineages, including the germline, maintaining genome integrity during the early stages of embryogenesis is potentially critical. Consistent with their adult roles, maternally inherited PIWI proteins have been observed in the pole plasm of syncytial embryos and in pole cells, the germ cell progenitors, after cellularization (*Brennecke et al., 2008*; *Dufourt et al., 2017*; *Mani et al., 2014*; *Megosh et al., 2006*).

Though, in adult flies, the piRNA pathway is restricted to the gonad, during the early phases of embryogenesis Piwi is also present in somatic nuclei (*Brennecke et al., 2008*; *Mani et al., 2014*; *Megosh et al., 2006*). This has long been taken as an indication that the piRNA pathway could play roles also in the developing soma, for example, helping to establish its epigenetic landscape (*Gu and Elgin, 2013*; *Seller et al., 2019*; *Yuan and O'Farrell, 2016*). However, probing piRNA pathway function during early embryogenesis has been hampered by a lack of suitable experimental approaches. Disrupting Piwi or other piRNA pathway factors in the female parent either via mutation or RNAi leads to oogenesis defects and often results in sterility or patterning defects that would confound the outcome of analyses (*Cox et al., 1998*; *Czech et al., 2013*; *Handler et al., 2013*; *Khurana et al., 2010*; *Klattenhoff et al., 2007*; *Klenov et al., 2011*; *Li et al., 2009a*; *Malone et al., 2009*; *Mani et al., 2014*; *Muerdter et al., 2013*; *Pane et al., 2007*; *Park et al., 2019*). RNAi-mediated depletion in embryos or generation of homozygous mutant embryos carrying piRNA pathway defects enables analysis of later developmental stages (*Akkouche et al., 2017*; *Gu and Elgin, 2013*; *Marie et al., 2017*), but not time windows where maternally deposited proteins predominate and generally drive development.

Here, we exploit a conditional protein degradation strategy to explore the function of maternally deposited piRNAs during *Drosophila* embryonic development. We find that Piwi-piRNA complexes present in the embryo are primarily derived from the oocyte, whereas components of the PICTS complex are both maternally deposited and zygotically expressed. An embryonic burst of transposon expression in somatic cells as the zygotic genome becomes active precedes the transient decoration of normally active elements in repressive chromatin marks. Rapid and efficient degradation of maternally deposited Piwi protein in embryos leads to earlier and increased activity of zygotically expressed TEs in concert with loss of repressive marks during the affected developmental stages. Although loss of transposon control in the embryonic soma does not result in an overt morphological phenotype, our results suggest that the piRNA pathway indeed plays a role in regulating the somatic chromatin structure during early embryogenesis. Through these mechanisms, a wave of expression, primarily of the *roo* transposon, is attenuated, though substantial expression of the TE remains.

## Results

### A transient burst of transposon expression during *Drosophila* embryogenesis

The maternal deposition of Piwi, Aub, and Ago3, noted more than a decade ago (*Brennecke et al., 2007*; *Brennecke et al., 2008*; *Gunawardane et al., 2007*; *Mani et al., 2014*; *Megosh et al., 2006*; *Rouget et al., 2010*), has long suggested a possible role for the piRNA pathway during embryogenesis. Prior studies have indicated that maternal instructions transmitted via piRNAs are important for defining piRNA clusters in the subsequent generation and/or provide critical information for gaining control over at least some transposons in daughters (*Akkouche et al., 2013*; *Akkouche et al., 2017*; *Brennecke et al., 2008*; *de Vanssay et al., 2012*; *Hermant et al., 2015*; *Khurana et al., 2011*; *Le Thomas et al., 2014a*; *Le Thomas et al., 2014b*). Both of these functions are relevant in gonadal cells. Yet, prior studies highlighted the presence of maternally deposited Piwi protein in the somatic nuclei of developing embryos (*Brennecke et al., 2008*; *Mani et al., 2014*; *Megosh et al., 2006*), leading to suggestions that piRNAs might help set the global epigenetic landscape of the embryonic soma (*Gu and Elgin, 2013*). To investigate the role of the piRNA pathway during embryogenesis, we first focused on its most well-established role, that of transposon control. Toward this end, we first characterized the expression of transposons throughout *Drosophila* embryogenesis by RNA-seq and quantitative mass spectrometry (*Figure 1A*).

Transcriptomes of 0–2 hr after egg laying (AEL) embryos represent the maternally inherited mRNA pool. Maternal transcripts are cleared and the zygotic genome is activated (zygotic genome activation [ZGA]) around nuclear cycle 14 (NC14; 2–2.5 hr AEL), and we generated RNA-seq data spanning 1 hr intervals of development from this point up to 10 hr AEL (stage 13). For comparison, we also included two late-stage embryo time points (12–13 hr and 17–18 hr AEL) as these were times when our prior data indicated that maternal Piwi was no longer detectable in somatic nuclei (*Brennecke et al., 2008*). To take into account different library sizes and facilitate comparability throughout our time-course experiment that only contained two biological replicates per time point,

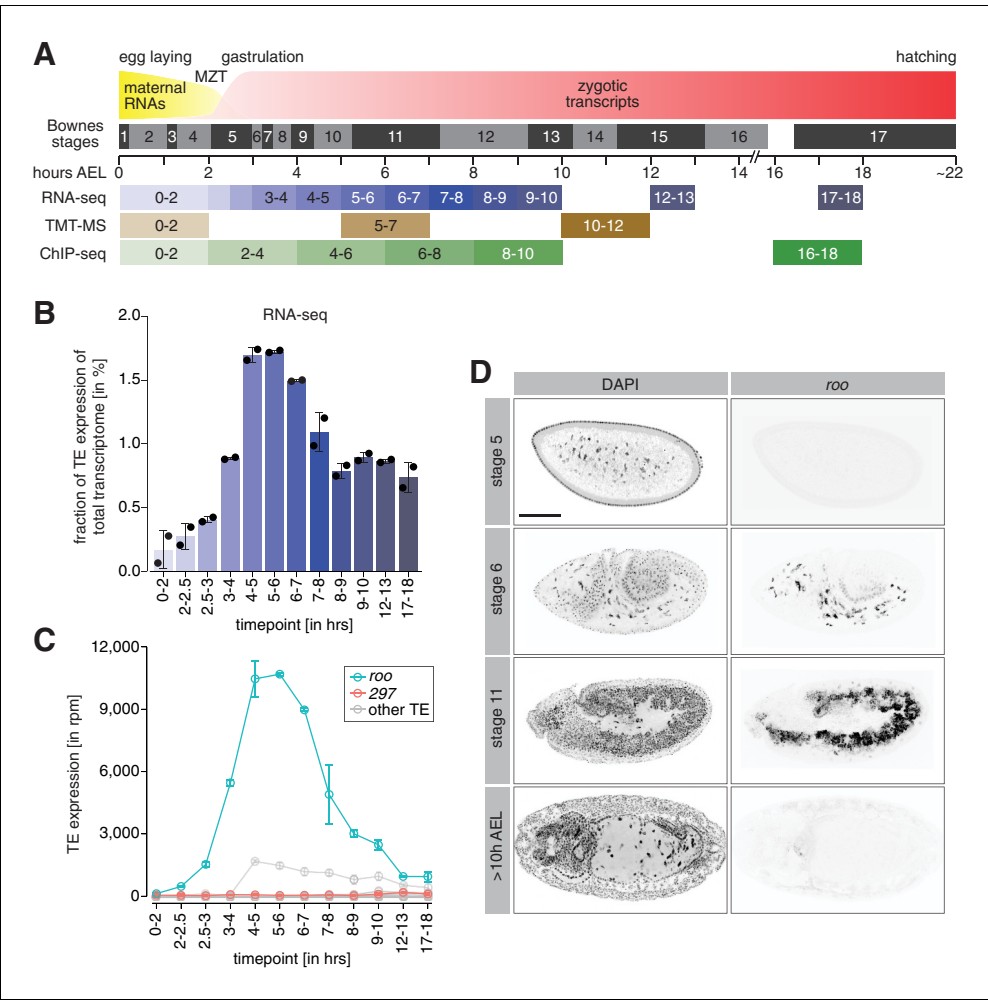

**Figure 1.** A transient burst of transposon expression during *Drosophila* embryogenesis. (**A**) Schematic of *Drosophila* embryogenesis indicating Bownes stages and collected time points. (**B**) Bar graphs showing contribution of transposon derived reads to the transcriptome of control *w1118* embryos at the indicated time points in percent. Error bars show standard deviation (n = 2). (**C**) Line graphs showing the RNA expression (in rpm) for the 30 most expressed transposons during the indicated time points of embryogenesis. Error bars show standard deviation (n = 2). (**D**) Confocal fluorescent microscopy images of control *w1118* embryos showing nuclei stained with DAPI and *roo* transcripts by RNA-FISH at the indicated embryonic stages (also see *Figure 1—figure supplement 2B*). Scale bar = 100 μm.

The online version of this article includes the following source data and figure supplement(s) for figure 1:

**Figure supplement 1.** Correlation of embryo collection time points between datasets.

**Figure supplement 1—source data 1.** Gene- and transposable element-mapping reads per million across embryogenesis RNA-seq time points for plots shown in *Figure 1B, C*, *Figure 1—figure supplement 1A, B*, *Figure 1—figure supplement 2A, F*, *Figure 2A*, and *Figure 2—figure supplement 1H, J*.

**Figure supplement 1—source data 2.** Signal intensity of genes and transposon open reading frames in TMT-MS time-course experiment for plots shown in *Figure 1—figure supplement 2A, C, D, F*, *Figure 2B*, and *Figure 2—figure supplement 1I, K*.

**Figure supplement 2.** Transposon protein and the known cohort of piRNA co-transcriptional gene silencing (coTGS) factors are present during embryogenesis.

our RNA-seq data was normalized to reads per million (rpm). We benchmarked our dataset by comparing the expression of selected embryonic genes in our RNA-seq to reported transcriptomes in FlyBase (*Graveley et al., 2011*). We found highly similar expression patterns of genes that are dynamically regulated during embryogenesis (*Figure 1—figure supplement 1A, B*). Furthermore, well-validated maternal (e.g., *fs(1)N* and *gammaTub37C*) and zygotic (e.g., *Ultrabithorax* [*Ubx*] and

*wingless* [*wg*]) genes demonstrated their expected expression patterns in our datasets (*Figure 1—figure supplement 2A*).

We detected only very few transposon transcripts in pre-ZGA embryos (0–2 hr AEL), as might be expected from their effective silencing by the piRNA pathway in ovaries. TE expression steadily increased following ZGA and peaked between 4 and 6 hr AEL (*Figure 1B*), similar to what was noted in prior reports (*Batut et al., 2013*). At the peak, transposon RNAs correspond to ~1.7% of the total embryonic transcriptome, with levels at the later studied time points decreasing to below 1% of the overall transcriptome. Transposons often show highly dynamic spatio-temporal expression; thus, we analyzed the contribution of individual TE families to the embryonic transcriptome. Interestingly, the majority of transposon expression could be attributed to one single transposon family, *roo* (*Figure 1C*). At its peak at 4–6 hr AEL, reads derived from the *roo* TE accounted for more than 1% of the entire embryonic transcriptome, corresponding to more than 70% of all TE-derived reads overall. From its expression peak, *roo* mRNA levels declined strongly before leveling off at around 12 hr AEL. While less pronounced, other transposons, such as *copia* and *297*, also showed dynamic expression changes during embryogenesis (*Figure 1—figure supplement 2F*).

The *roo* expression peak at 4–6 hr AEL could be due to transcription from germ cell precursors, which become transcriptionally active around 3.5 hr AEL (stage 8) (*Van Doren et al., 1998*; *Zalokar, 1976*). However, the sheer abundance of *roo* and other transposon transcripts argued strongly that they must emanate at least in part from somatic nuclei as these vastly outnumber the germ cell precursors. To directly test the origin of *roo* transcripts during embryogenesis, we performed RNA fluorescence in situ hybridization (RNA-FISH). In agreement with our RNA-seq data, *roo* transcripts were detected as early as stage 6 (in gastrulating embryos ~3 hr AEL) and localized predominantly to yolk cell nuclei (*Figure 1D*, *Figure 1—figure supplement 2B*). Stage 11 embryos (~5 hr AEL) showed strong *roo* RNA signal in somatic cells of the mesoderm, similar to earlier reports (*Brönner et al., 1995*; *Ding and Lipshitz, 1994*). In contrast, *roo* transcript levels were undetectable by FISH in late-stage embryos (>10 hr AEL). These data indicate a transient somatic burst of *roo* expression during early *Drosophila* development.

TEs rely on proteins encoded in their open reading frames (ORFs) for mobilization. *roo* is an LTR retrotransposon and, as has been proposed for *gypsy* in ovarian follicle cells (*Kim et al., 1994*; *Leblanc et al., 2000*; *Song et al., 1997*), could potentially be packaged into virion-like particles, possibly enabling infection of germ cell precursors as a propagation mechanism. To determine whether *roo*-encoded proteins are expressed in embryos, we mined quantitative proteomic data from three developmental intervals (*Figure 1A*). The first, 0–2 hr AEL, represents the time before ZGA when the proteome is derived from maternal protein deposition and zygotic translation of maternal mRNAs. The second, 5–7 hr AEL, represents an interval where zygotic *roo* expression had become robust, and the third, 10–12 hr AEL, is a time at which *roo* RNA levels had substantially declined.

In transcriptionally silent embryos (0–2 hr AEL), we detected over 6400 unique proteins. Compared to 0–2 hr embryos, the signal intensity of ~17% or 1114 proteins significantly increased (p<0.01) by over 25% in 5–7 hr AEL embryos (*Figure 1—figure supplement 2C*). We also detected 490 (or ~8% of) proteins that significantly decreased (p<0.01) by over 25% in 5–7 hr AEL embryos (*Figure 1—figure supplement 2D*). The majority of proteins (4652 or 72%), however, did not change by more than 25% between 0–2 hr and 5–7 hr AEL embryos. As with transcriptome analyses, known maternally deposited and zygotically expressed proteins showed their expected patterns of presence in the datasets.

Compared to the early time point (0–2 hr AEL), 5–7 hr AEL embryos showed significant accumulation of *roo* peptides (p<0.01) corresponding to its expression peak. *roo* encodes a single ORF (with a predicted protein weight of 272 kDa), which contains a group-specific antigen-like protein (gag), a reverse transcriptase (RT/pol), an envelope protein (env), two peptidases-like domains (Pep), and a zinc finger (*Figure 1—figure supplement 2E*). We detected peptides corresponding to the gag, pol, and env proteins (*Figure 1—figure supplement 2E*, bottom), indicating potential competence for retrotransposition. We additionally detected proteins derived from other transposons including *copia* and *297*. Of note, *roo* ORFs remained detectable at 10–12 hr AEL (*Figure 1—figure supplement 2F*), possibly suggesting substantial stability, as this was a time at which *roo* mRNA levels had diminished.

## The known cohort of piRNA coTGS factors is present during embryogenesis

The decline in transposon expression from 4 to 6 hr to 10–12 hr intervals of embryogenesis could potentially involve the piRNA pathway. However, piRNA-guided post-transcriptional or co-transcriptional silencing also requires a growing list of additional proteins (reviewed in *Czech et al., 2018*; *Ozata et al., 2019*). We therefore probed the expression of known piRNA pathway components during various stages of embryogenesis in our transcriptomic and proteomic datasets.

With the exception of Piwi, genes involved in coTGS were both maternally deposited and zygotically expressed during the first ~10 hr of embryogenesis (*Figure 1—figure supplement 2A*). Components of the PICTS complex, comprising Panx, Nxf2, Nxt1, and Ctp, showed abundant protein expression in the 5–7 hr and 10–12 hr AEL time intervals. piRNA-mediated coTGS also depends on several general chromatin modifiers, including Egg and its co-factor Wde (*Osumi et al., 2019*; *Rangan et al., 2011*; *Sienski et al., 2015*; *Yu et al., 2015*). Both of these proteins are required for heterochromatin formation in the embryo, and Egg in particular has previously been implicated in embryonic repeat silencing (*Seller et al., 2019*). Similar to piRNA-specific coTGS factors, proteins involved in general chromatin silencing were both maternally deposited and zygotically expressed and detected at all studied time points, as expected based on their ubiquitous functions (*Figure 1—figure supplement 2A*). Of note, Piwi mRNA appears to be primarily maternally deposited, with zygotic transcript levels remaining low throughout embryogenesis (*Figure 1—figure supplement 2A*, *Figure 2A, B*).

In contrast, we noted little or no maternal deposition and low zygotic expression of key components of the piRNA precursor expression and export machinery and of critical piRNA biogenesis factors (*Figure 1—figure supplement 2A*). Considered together, our expression analyses are consistent with the potential of maternally instructed Piwi protein acting through coTGS during *Drosophila* embryogenesis.

## Components of the piRNA-guided coTGS machinery are enriched in somatic and pole cell nuclei during embryogenesis

To assess the potential role of the piRNA pathway in regulating the transposon burst during *Drosophila* embryogenesis, we examined the spatial and temporal expression of coTGS proteins in the developing embryo using light-sheet live fluorescence microscopy. For this purpose, we used two previously published lines carrying a modified BAC expressing either GFP-Piwi or GFP-Panx from its endogenous regulatory region (*Handler et al., 2013*; *Sienski et al., 2015*) and a GFP-Nxf2 knock-in line that we generated by CRISPR/Cas9 (*Fabry et al., 2019*). As Ctp and Nxt1 have many additional functions, we did not examine their localization in this study. We also crossed in a transgene carrying H2Av-RFP to enable tracking of nuclei. Pre-blastoderm stage embryos (0.5 hr AEL) were continuously imaged for >10 hr of embryogenesis. As previously reported (*Brennecke et al., 2008*; *Mani et al., 2014*; *Megosh et al., 2006*) and consistent with its maternal deposition, we detected GFP-Piwi during the pre-blastoderm stage (NC1–9, ~0–30 min AEL) localized to the posterior pole where it formed a crescent-like structure (*Video 1*, *Figure 2C*).

As embryogenesis progressed and somatic nuclei migrated to the surface (NC 9–14, ~1.5–3 hr AEL), Piwi localized to somatic nuclei and to the pole plasm surrounding the nuclei of germline progenitor cells, as we and others reported earlier based on immunofluorescence staining of fixed embryos (*Brennecke et al., 2008*; *Mani et al., 2014*; *Megosh et al., 2006*). In agreement with an earlier report (*Mani et al., 2014*), our dynamic data revealed that nuclear Piwi signal strongly decreased during mitotic cycles, with little fluorescence signal overlapping with H2Av-RFP during nuclear divisions (*Video 1*, *Figure 2D*). We continued to detect Piwi expression in somatic nuclei throughout the first 10 hr of embryogenesis; however, signal intensity decreased over time. This observation was consistent with transcriptomic and proteomic measurements taken over a comparable time course (*Figure 2A, B*).

Similar to Piwi, both Nxf2 and Panx were detected in somatic and pole cell nuclei from the syncytial blastoderm stage (*Videos 2* and *3*, *Figure 2—figure supplement 1A–E*). In contrast to Piwi, Panx and Nxf2 showed strong co-localization with H2Av-RFP during mitotic cycles (*Videos 2* and *3*, *Figure 2—figure supplement 1F, G*), suggesting that while Piwi is predominantly excluded, Nxf2 and Panx are retained in the nucleoplasm during mitosis. Consistent with our RNA-seq and TMT-MS

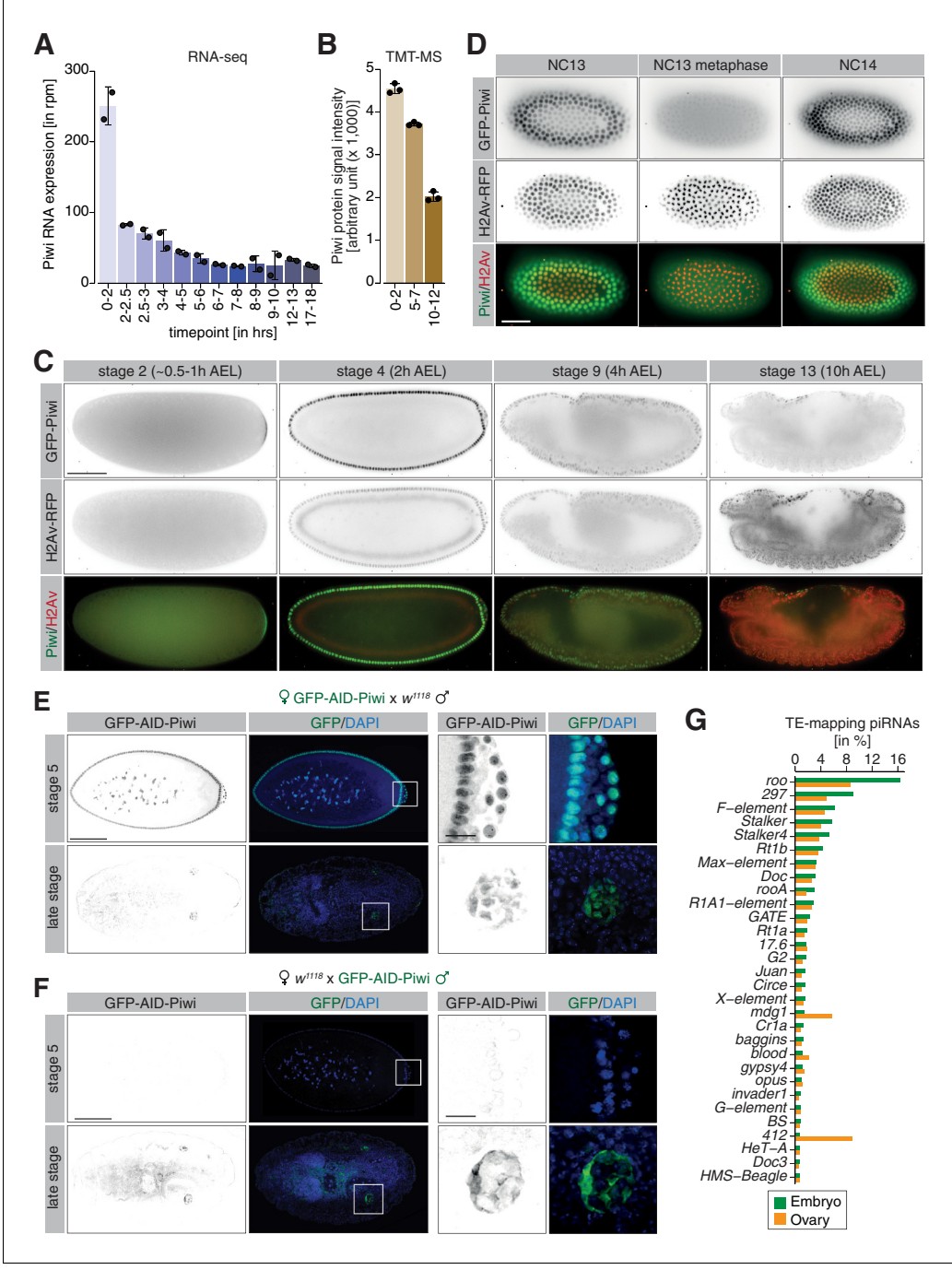

**Figure 2.** piRNA co-transcriptional gene silencing (coTGS) factors are maternally inherited and localize to somatic cells of the *Drosophila* embryo. (A) Bar graphs showing Piwi RNA expression (in rpm) at the indicated time points in control $w^{1118}$ embryos. Error bars show standard deviation (n = 2). (B) Bar graphs showing Piwi protein signal intensity (arbitrary units) at the indicated time points in control $w^{1118}$ embryos. Error bars show standard deviation (n = 3). (C) Stand-still images from **Video 1** obtained by light-sheet fluorescent live microscopy of embryos derived from parents expressing GFP-Piwi (green) and H2Av-RFP (red) for the indicated time points. Scale bar = 50 μm. (D) As in (C) but showing the transition from NC13 to NC14. Scale bar = 50 μm. (E) Confocal fluorescent microscopy images of embryos derived from females expressing GFP-AID-Piwi crossed to control $w^{1118}$ males probing for GFP and DAPI. Shown are embryos at the blastoderm stage (stage 5) and late-stage embryos (>12 hr after egg laying). Scale bar = 100 μm. Zoom of the indicated regions showing developing germ cells. Scale bar = 10μm. (F) As in (E) but showing embryos derived from control $w^{1118}$ females crossed to GFP-AID-Piwi males. (G) Bar graph showing small RNA-seq from Piwi immunoprecipitation of 0–8 hr control $w^{1118}$ embryos (green, n = 1) or adult ovaries

*Figure 2 continued on next page*

*Figure 2 continued*

(orange, n = 1). Shown are antisense piRNAs of the 30 most abundant TE families in embryos as percentage of reads mapping to indicated transposons relative to all transposable element-mapping antisense piRNAs.

The online version of this article includes the following figure supplement(s) for figure 2:

**Figure supplement 1.** Maternally inherited co-transcriptional gene silencing (coTGS) factors localize to nuclei of pole and somatic cells during embryogenesis.

data (*Figure 2—figure supplement 1H–K*), as embryogenesis progressed, Panx and Nxf2 remain detectable for several hours (>10 hr AEL), closely matching the protein expression of Piwi.

Piwi carries epigenetic information in the form of piRNAs (*Brennecke et al., 2008*; *Le Thomas et al., 2014b*). However, it is unclear if Piwi-piRNA complexes are assembled during oogenesis prior to maternal deposition into the embryo, or whether zygotic piRNA biogenesis and Piwi loading also occurs. We therefore analyzed the expression of GFP-tagged Piwi from reciprocal crosses with control *w1118* flies by immunofluorescence staining in early and late-stage embryos. Embryos derived from females expressing GFP-Piwi showed strong maternal deposition of Piwi during early embryogenesis (*Figure 2E*, *Figure 2—figure supplement 1L*), with GFP fluorescence in later stage (>12 hr AEL) embryos restricted to the germline cells. Consistent with maternal deposition of Piwi, embryos derived from the reciprocal cross showed no GFP signal in the early embryos (*Figure 2F*, *Figure 2—figure supplement 1M*). Instead, we only observed GFP-Piwi signal in the developing gonads of late-stage embryos, likely as a result of zygotic expression. Strikingly, Piwi of zygotic origin localized exclusively to the cytoplasm of the germ cell progenitors and was not detected in nuclei, suggesting that zygotically transcribed Piwi is likely not relevant for coTGS until later in development.

## The decline in embryonic transposon expression is correlated with hallmarks of piRNA-dependent co-transcriptional silencing

Piwi proteins are guided by their piRNA co-factors to recognize and co-transcriptionally silence active transposons in the *Drosophila* ovary (*Le Thomas et al., 2013*; *Post et al., 2014*; *Sienski et al., 2015*; *Sienski et al., 2012*; *Yu et al., 2015*). If this pathway were relevant in the embryonic soma, maternally deposited Piwi would require instructions to recognize embryonically expressed elements. To examine this possibility, we immunoprecipitated Piwi from 0 to 8 hr control *w1118* embryos as well as from adult ovaries and sequenced the associated small RNAs. Piwi in both tissues existed in complex with 23- to 28-nt piRNAs and showed nearly indistinguishable size profiles that were biased for antisense reads (*Figure 2—figure supplement 1N, O*). Closer inspection by aligning the reads to transposon consensus sequences revealed similar piRNA levels for the majority of TEs; however, we detected some notable differences (*Figure 2G*). Piwi in ovaries showed higher levels of antisense piRNAs targeting the TEs *mdg1* and *412*, in agreement with the majority of these

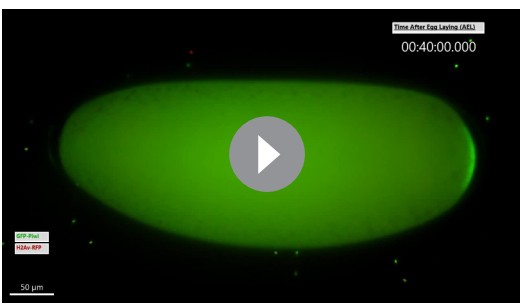

**Video 1.** GFP-Piwi live imaging. Light-sheet fluorescent live microscopy of embryos derived from parents expressing GFP-Piwi (green) and H2Av-RFP (red) in developing embryos for indicated time after egg laying.

https://elifesciences.org/articles/68573#video1

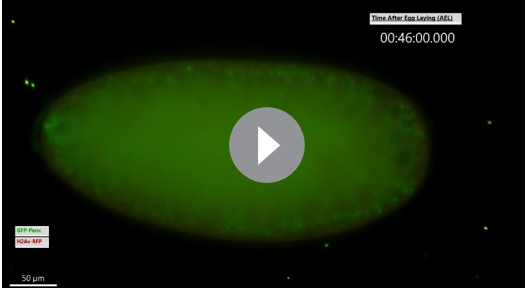

**Video 2.** GFP-Panx live imaging. Light-sheet fluorescent live microscopy of embryos derived from parents expressing GFP-Panx (green) and H2Av-RFP (red) in developing embryos for indicated time after egg laying.

https://elifesciences.org/articles/68573#video2

small RNAs originating from the soma-specific *flam* piRNA cluster (*Brennecke et al., 2007*; *Malone et al., 2009*; *Zanni et al., 2013*). Piwi in embryos showed high levels of antisense piRNAs targeting *roo* (~16% or all TE-targeting reads) and *297* (~9%), consistent with an ability of maternally deposited Piwi to potentially recognize these TEs when expressed in the embryo.

In *Drosophila* ovaries, coTGS depends on Piwi-mediated recruitment of the PICTS/SFiNX complex and correlates with the deposition of H3K9me3 marks at TE insertions and surrounding genomic regions (*Batki et al., 2019*; *Eastwood et al., 2021*; *Fabry et al., 2019*; *Murano et al., 2019*; *Schnabl et al., 2021*; *Sienski et al., 2015*; *Yu et al., 2015*; *Zhao et al., 2019*). Due to the poor conservation of the genomic locations of transposon insertions between different *Drosophila* strains, we used whole-genome sequencing (WGS) to de novo identify the TE insertion sites present in our control *w1118* flies (see Materials and methods). This data enabled us to identify over 600 euchromatic transposon insertions that are absent from the dm6 reference genome, and these were used for our chromatin analyses, as most annotated insertions in the dm6 genome assembly were absent from our strain.

In order to determine the fate of transposon loci throughout embryogenesis, we performed H3K9me3 chromatin immunoprecipitation followed by sequencing (ChIP-seq) on control *w1118* embryos at 2 hr intervals covering the period when transposon expression is dynamic (0–10 hr AEL) and a later time point (16–18 hr AEL) well after maternal Piwi protein was no longer detectable in somatic nuclei (*Figures 1A* and *2C*). We included adult ovaries, which show piRNA-guided coTGS, as well as adult heads, a somatic tissue without active piRNA pathway, to compare the changes of this repressive chromatin mark across different stages and tissues of *Drosophila* development.

Early embryos (0–2 hr AEL) showed low levels of H3K9me3 signal at 117 euchromatic, *w1118*-specific *roo* insertions (*Figure 3A*). However, as development progressed, H3K9me3 levels increased with a peak at 6–10 hr AEL (*Figure 3A*, *Figure 3—figure supplement 1A*). Thus, deposition of repressive chromatin marks correlated with the RNA expression of *roo*, yet the maximum of H3K9me3 accumulation lagged behind the RNA expression peak by approximately 2 hr. These data are consistent with a requirement for nascent transcription for efficient conversion of a TE insertion into heterochromatin, as previously reported in yeast (*Bühler et al., 2006*; *Shimada et al., 2016*) and for the recognition of transposon loci by the piRNA pathway (*Le Thomas et al., 2013*; *Post et al., 2014*; *Rozhkov et al., 2013*; *Sienski et al., 2015*; *Sienski et al., 2012*; *Yu et al., 2015*). Of note, the deposition of repressive marks trailed the direction of transcription and showed higher signal enrichments in the regions downstream of the transposon insertions, as previously observed for piRNA-dependent silencing in cell culture systems (*Fabry et al., 2019*; *Sienski et al., 2015*; *Sienski et al., 2012*).

Interestingly, H3K9me3 signal at euchromatic *roo* insertions of 16–18 hr AEL embryos, which lacked maternal Piwi in somatic nuclei and no longer express *roo*, showed diminished intensities compared to earlier time intervals. Similarly, heads and ovaries, both tissues from adult flies, showed no enrichment of H3K9me3 at euchromatic *roo* insertions, despite the presence of a functional piRNA pathway in ovaries. Considered together, these data suggest that maternal piRNAs program a response to a burst of *roo* expression during embryonic development but that the deposition of H3K9me3 marks, likely directed via coTGS, no longer occurs at developmental time points and in tissues where *roo* is not expressed. This is consistent both with the known requirement for active transcription for targeting by Piwi and with the observed need for continuous engagement of the PICTS/SFiNX complex to maintain H3K9me3 marks on transposon loci (*Batki et al., 2019*; *Eastwood et al., 2021*; *Fabry et al., 2019*; *Le Thomas et al., 2013*; *Murano et al., 2019*; *Post et al., 2014*; *Rozhkov et al., 2013*; *Schnabl et al., 2021*; *Sienski et al., 2015*; *Sienski et al., 2012*; *Yu et al., 2015*; *Zhao et al., 2019*).

To investigate whether this mechanism is specific to *roo* or more general, we examined the transposon *297*, which is also expressed during embryogenesis (*Figure 1—figure supplement 1F*) and showed high targeting potential by maternally inherited piRNAs (*Figure 2G*). Genomic loci in close proximity to euchromatic, *w1118*-specific *297* insertions (n = 20) showed dynamic deposition of H3K9me3 similar to *roo* (*Figure 3—figure supplement 1B*). However, while H3K9me3 levels at *roo* insertions peaked between 6 and 10 hr AEL, *297* insertions showed the maximum H3K9me3 signal intensity between 2 and 8 hr AEL, suggesting that these loci are targeted by coTGS earlier than *roo* insertions. In contrast, H3K9me3 occupancy at transposons such as *mdg1* and *412* that were expressed during embryogenesis but lacked substantial maternal deposition of piRNAs retained low

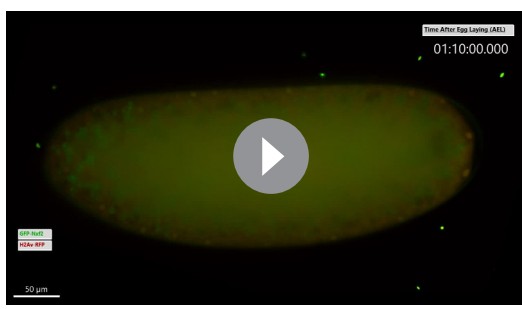

**Video 3.** GFP-Nxf2 live imaging. Light-sheet fluorescent live microscopy of embryos derived from parents expressing GFP-Nxf2 (green) and H2Av-RFP (red) in developing embryos for indicated time after egg laying.
https://elifesciences.org/articles/68573#video3

H3K9me3 levels throughout embryogenesis, though they showed a strong enrichment in ovaries (*Figure 3—figure supplement 1C*).

To determine whether the deposition of repressive chromatin marks at euchromatic *297* and *roo* insertions was specific, rather than reflecting a general trend of H3K9me3 accumulation genome-wide, we analyzed genomic regions not targeted by maternally inherited piRNAs. H3K9me3 signal at constitutive heterochromatin remained stable throughout the sampled time points (*Figure 3—figure supplement 1D*), while H3K9me3 levels on chromosome 4 increased steadily throughout development (*Figure 3—figure supplement 1E*). Of note, while ovaries showed no coTGS signature at *roo* insertions, other transposons, such as *Doc*, showed a clear accumulation of H3K9me3 marks that was absent in embryos during all assayed time points (*Figure 3B*). Considered together, these results are consistent with piRNA-guided chromatin modification of a subset of transposons that show activity during *Drosophila* embryonic development.

## An auxin-inducible degron enables rapid depletion of Piwi in ovaries and early embryos

Though embryonically repressed transposons bore hallmarks of piRNA-guided heterochromatin formation, the reliance of the pathway on maternally deposited Piwi-piRNA complexes prevented a demonstration that silencing depended on the pathway through conventional genetics. Ovaries that lack key piRNA pathway silencing factors show substantial expression changes and produce morphologically altered eggs that largely fail to develop normally (*Cox et al., 1998*; *Czech et al., 2013*; *Handler et al., 2013*; *Khurana et al., 2010*; *Klattenhoff et al., 2007*; *Klenov et al., 2011*; *Li et al., 2009a*; *Malone et al., 2009*; *Mani et al., 2014*; *Muerdter et al., 2013*; *Pane et al., 2007*; *Park et al., 2019*).

To investigate the effect of Piwi depletion on *Drosophila* embryogenesis without affecting oogenesis, we used the auxin-inducible degron (AID) system (*Nishimura et al., 2009*). This protein degradation system comprised an AID-tag, fused to the protein of interest, and the plant-derived F-box protein transport inhibitor response 1 (TIR1). AID and TIR1 associate with each other in an auxin-dependent manner, with binding of TIR1 to the AID-tagged target leading to the recruitment of the cellular ubiquitination machinery and target protein degradation via the proteasome (*Figure 4A*). This conditional degradation system has proven effective in several model organisms including *Drosophila* where it was recently shown to enable degradation of the germ cell-specific protein Vasa (*Bence et al., 2017*).

We used CRISPR/Cas9 to insert an amino-terminal GFP-AID tag at the *Drosophila piwi* locus and crossed these flies to a line expressing the *Oryza sativa*-derived TIR1 (OsTIR1) protein under the control of the *ubiquitin* promoter. As a proof of concept, we tested the auxin-induced degradation of Piwi in adult ovaries of flies homozygous for both GFP-AID-Piwi and OsTIR1. Feeding flies for 24 hr with 5 mM auxin-containing yeast paste was sufficient to induce complete degradation of Piwi in ovaries (*Figure 4B, C*, *Figure 4—figure supplement 1A*), and this depletion resulted in the derepression of transposons (*Figure 4D*). Notably stronger changes were observed following longer treatments, possibly implying a lag between loss of piRNA pathway function and that of repressive chromatin marks. Following a 1-day treatment, embryos laid by Piwi-depleted females developed without defects and showed similar hatching rates as their control treated siblings (*Figure 4E, F*). Longer auxin treatments resulted in more frequent deformation of embryos that was accompanied by reduced hatching rates (*Figure 4E, F*), likely due to patterning defects as a result of Piwi depletion from follicle cells.

*Drosophila* embryos develop within a relatively impermeable chorion, and treatment of embryos directly with auxin showed little impact. However, in dechorionated embryos we observed a near

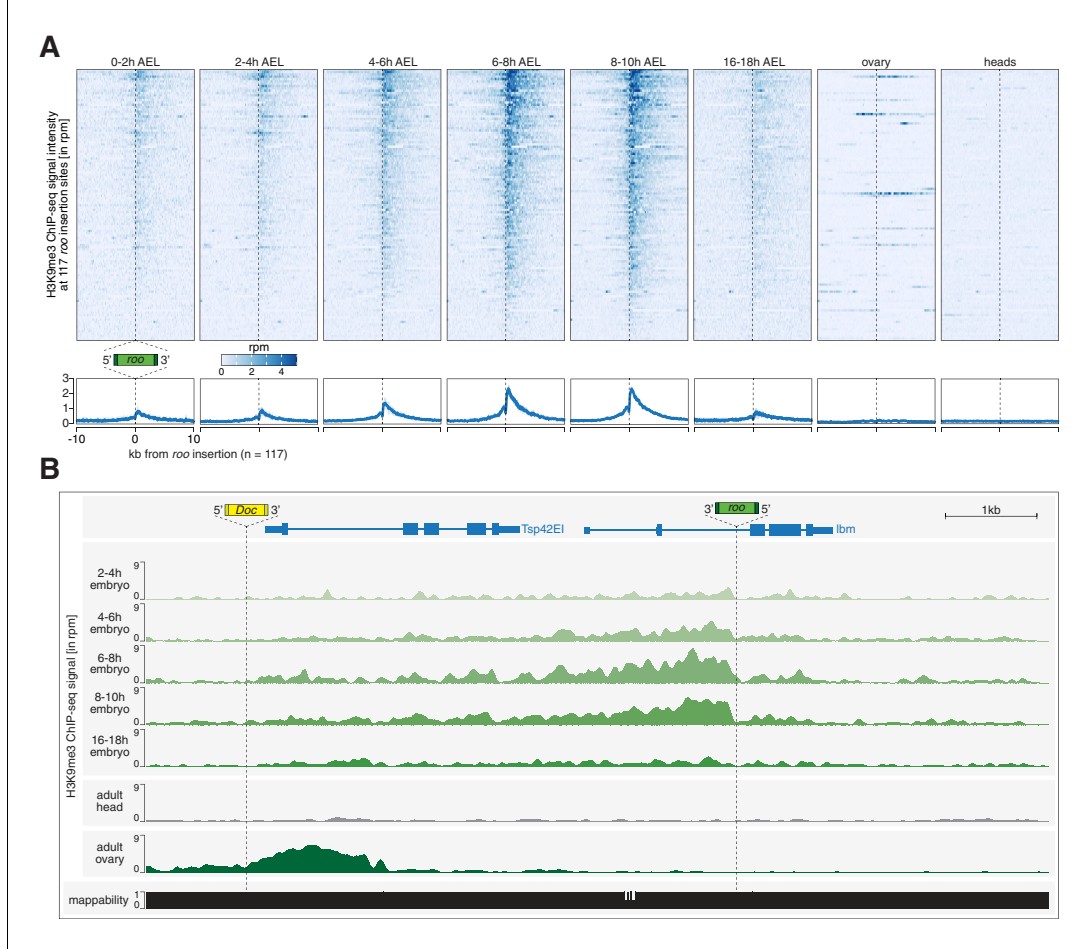

**Figure 3.** Transposon insertions targeted by piRNAs in embryos show epigenetic changes characteristic of co-transcriptional gene silencing. (**A**) Heatmaps (top) and metaplots (bottom) showing H3K9me3 ChIP-seq signal (in rpm) for the indicated embryonic stages and adult tissues at 117 euchromatic, $w^{1118}$-specific *roo* insertions (n = 2). Signal is shown within 10 kb from insertion site and sorted from 5′ to 3′. (**B**) UCSC genome browser screenshot showing H3K9me3 ChIP-seq signal for the indicated genes on chromosome 2R carrying $w^{1118}$-specific *roo* and *Doc* insertions.

The online version of this article includes the following figure supplement(s) for figure 3:

**Figure supplement 1.** Transposons targeted by maternally inherited piRNAs show hallmarks of co-transcriptional gene silencing (coTGS).

complete degradation of Piwi protein following 30 min auxin treatment of embryos collected for 0–30 min AEL (*Figure 5A, B*). To investigate the dynamics of auxin-mediated Piwi depletion in embryos, we used light-sheet fluorescence live microscopy. Early blastoderm embryos treated with 5 mM auxin showed rapid degradation of GFP-AID-Piwi signal, which was undetectable after 25 min of treatment (*Figure 5C*, *Video 4*). Of note, the removal of maternal Piwi in this time window did not significantly affect the embryo hatching rate (*Figure 4—figure supplement 1B*).

## Maternally deposited Piwi directs heterochromatin formation at active transposon insertions during early embryogenesis

We next investigated the impact of degrading maternal Piwi from early-stage embryos on transposons. Embryos derived from flies homozygous for GFP-AID-Piwi and OsTIR1 were collected across a 30 min period and treated for an additional 2.5 hr with or without 5 mM auxin before RNA extraction, generation of libraries, and differential expression analysis of the sequenced transcriptomes (*Figure 5A*). These embryos corresponded to 2.5–3 hr AEL, the point at which we began to observe zygotic *roo* transcripts (*Figure 1C*), and showed minimal differences between control- and auxin-treated embryos for the same set of genes used to benchmark our dataset (*Figure 5—figure supplement 1A*). The majority of transposons showed no significant expression change upon Piwi

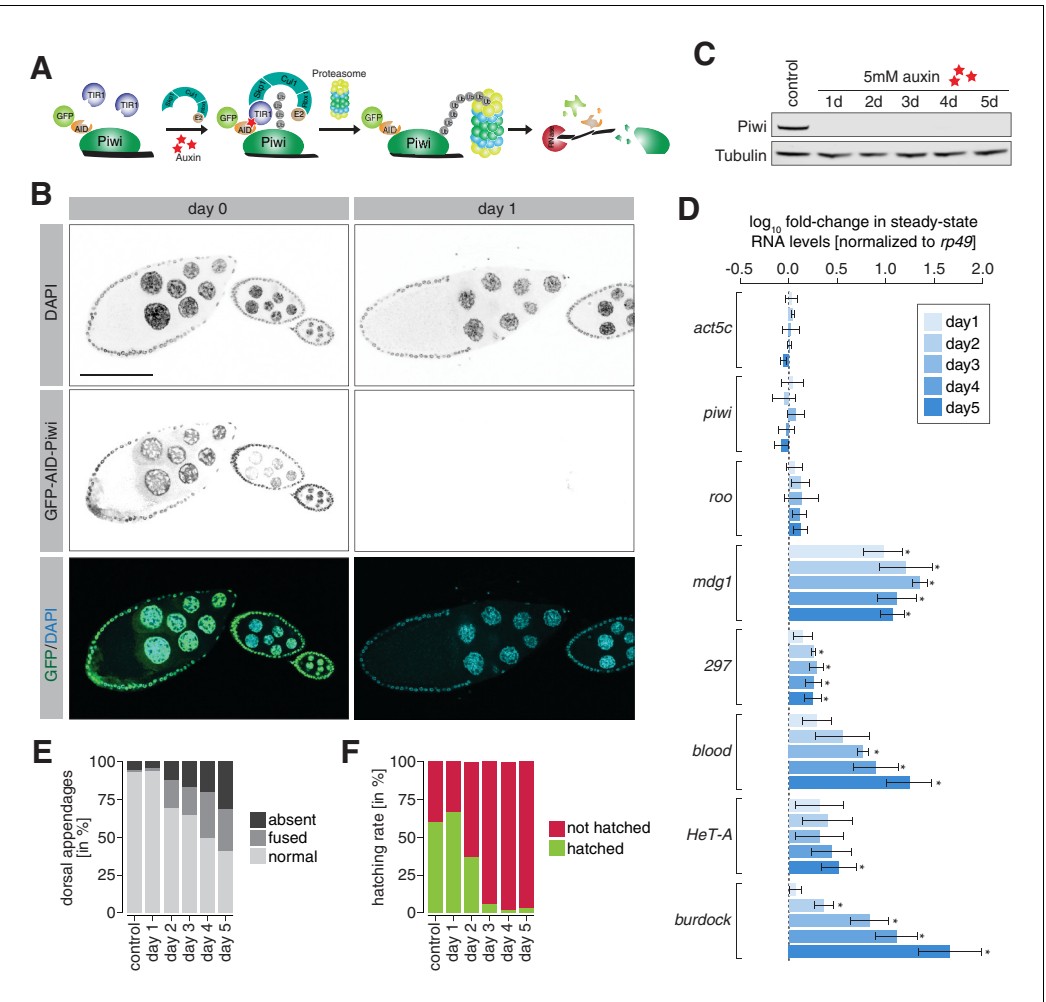

**Figure 4.** Degradation of Piwi protein in ovaries resembles mutant phenotypes. (**A**) Cartoon illustrating the Piwi protein degradation strategy using the auxin-inducible AID-TIR1 system. (**B**) Confocal fluorescent microscopy images showing ovary egg chambers of GFP-AID-Piwi; OsTIR1 flies fed with yeast paste containing 5 mM auxin for the indicated time (also see *Figure 4—figure supplement 1A*). Blue = DAPI. Green = GFP-AID-Piwi. (**C**) Western blot of ovaries from females treated with 5 mM auxin-containing yeast paste for the indicated time period or control females probing for Piwi and Tubulin as a loading control. (**D**) Bar graphs showing *rp49*-normalized steady-state RNA levels of the indicated transposable elements and control genes in ovaries of GFP-AID-Piwi; OsTIR1 flies fed with yeast paste containing 5 mM auxin for the indicated time. Error bars show standard deviation (n = 3). Asterisk denotes significant changes compared to control (p<0.05, unpaired t-test). (**E**) Bar graphs showing the percentage of embryo deformation phenotypes laid by GFP-AID-Piwi; OsTIR1 females fed with yeast paste containing 5 mM auxin for the indicated time. (**F**) As in (**E**) but showing the hatching rate in percent.
The online version of this article includes the following figure supplement(s) for figure 4:

**Figure supplement 1.** Piwi degradation in ovaries resembles knockdown and mutant phenotypes.

depletion; however, *roo* and *297* were significantly de-repressed (p<0.05) by more than twofold (*Figure 5D*), suggesting that Piwi impacts their expression during embryogenesis. Previous studies suggested that auxin in small concentrations has a negligible impact on *Drosophila* development (*Bence et al., 2017*; *Trost et al., 2016*), but to control for effects of auxin itself on TE regulation, we also evaluated transposon expression in auxin-treated GFP-AID-Piwi embryos that lack OsTIR1. Without OsTIR1, 2.5–3 hr embryos treated with 5 mM auxin showed no significant changes in transposon expression compared to control siblings treated with PBS (*Figure 5—figure supplement 1B*).

We additionally examined changes in the repressive chromatin mark H3K9me3 to determine whether these were deposited in a piRNA-dependent fashion at euchromatic *roo* and *297*

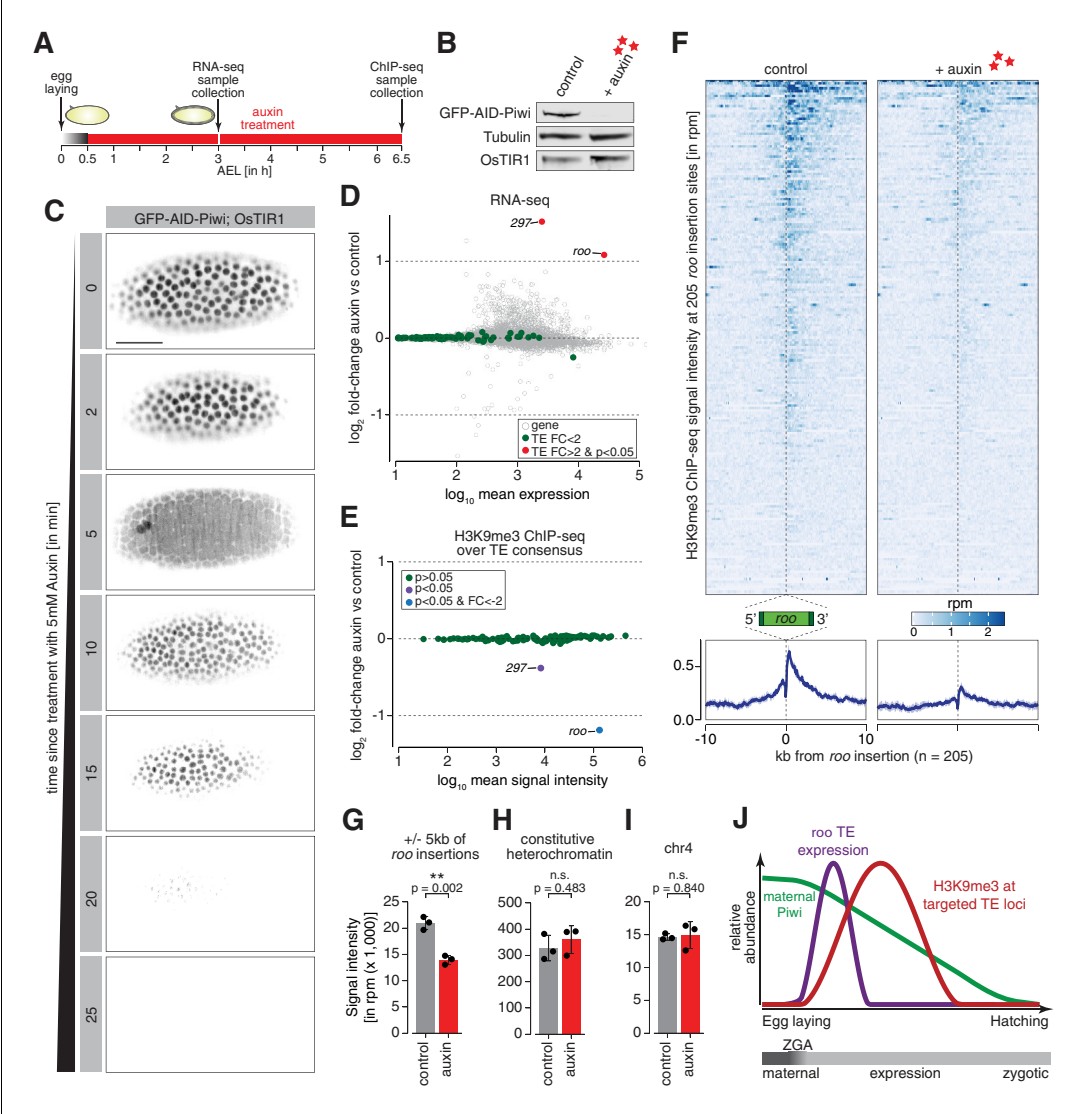

**Figure 5.** Degradation of maternally deposited Piwi in embryos leads to transposon deregulation. (A) Schematic of embryo auxin treatments and sample collection for RNA-seq and ChIP-seq experiments. (B) Western blot showing abundance of GFP-AID-Piwi fusion protein in embryos treated with 5 mM auxin for 2 hr. OsTIR1 and tubulin expression are shown as loading control. (C) Stand-still images from *Video 4* obtained by light-sheet fluorescent live microscopy of embryos derived from parents expressing GFP-AID-Piwi and OsTIR1 treated with 5 mM auxin for the indicated time intervals. Scale bar = 50 μm. (D) MA plot showing base mean expression ($\log_{10}$ scale) of transposon RNAs relative to their fold-change ($\log_2$ scale) in GFP-AID-Piwi; OsTIR1 embryos treated with 5 mM auxin versus control (n = 3). Gray = genes, green = TEs not changed (p<0.05), red = transposable elements (TEs) significantly changed (p<0.05) and fold-change > 2. (E) MA plot showing base mean signal intensity ($\log_{10}$ scale) of TEs relative to the H3K9me3 ChIP-seq signal enrichment ($\log_2$ scale) in GFP-AID-Piwi; OsTIR1 embryos treated with 5 mM auxin versus control (n = 3). Gray = TEs not significantly changed (p>0.05), purple = TEs significantly changed (p<0.05), blue = TEs significantly changed (p<0.05) and fold-change < –2. (F) Heatmaps (top) and metaplots (bottom) showing H3K9me3 ChIP-seq signal (in rpm) for control embryos and 5 mM auxin-treated embryos at 205 euchromatic, degron strain-specific *roo* insertions (n = 3). Signal is shown within 10 kb from insertion site and sorted from 5′ to 3′. (G) Bar graphs showing H3K9me3 signal intensity (in rpm) for the indicated treatments at *roo* loci. Error bars show standard deviation (n = 3). Statistics were calculated with unpaired (two-sample) t-test. (H) As in (G) but showing constitutive heterochromatin. (I) As in (G) but showing chromosome 4 regions. (J) Model of piRNA-guided chromatin modification at active transposons in somatic cells of the developing *Drosophila* embryo.

The online version of this article includes the following figure supplement(s) for figure 5:

**Figure supplement 1.** Piwi depletion in embryos leads to epigenetic changes at transposable element (TE) insertions targeted by maternally inherited piRNAs.

transposon insertions. We again collected embryos for 30 min and treated with 5 mM auxin (or PBS

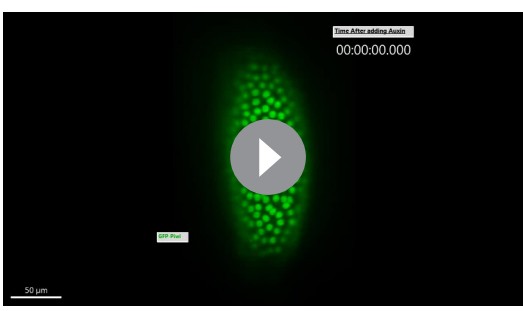

**Video 4.** GFP-AID-Piwi live imaging. Light-sheet fluorescent live microscopy of embryos derived from parents expressing GFP-AID-Piwi (green) and OsTIR1. Time course shows early embryo treated with 5 mM auxin for indicated time points.

https://elifesciences.org/articles/68573#video4

as a negative control) for 6 hr, which yielded embryos 6–6.5 hr AEL (*Figure 5A*) and corresponds to the peak in H3K9me3 signal at *roo* insertions in control $w^{1118}$ embryos (*Figure 3A*, *Figure 3—figure supplement 1A*). Piwi depletion severely impacted H3K9me3 signal over the transposon consensus sequence of *roo* and *297*, but not that of other TEs (*Figure 5E*). Additionally, H3K9me3 levels at individual transposon genomic loci (see Materials and methods for identification of TE insertions in our fly stock) showed similar patterns. H3K9me3 signal in genome-wide 5 kb bins predominantly changed when *roo* or *297* insertions were nearby (*Figure 5—figure supplement 1C*). We identified 154 bins with significantly reduced (p<0.05) H3K9me3 occupancy, while only two bins showed an increase. Of the bins with lower H3K9me3 signal, 122 and 10 were within 5 kb of *roo* or *297*

insertions, respectively, thereby illustrating the impact of Piwi on chromatin states at genomic loci specifically targeted by maternal piRNAs. Furthermore, 205 and 63 individual euchromatic, degron strain-specific TE insertions of both *roo* and *297*, respectively, showed a strong decrease of H3K9me3 levels in Piwi-depleted embryos (*Figure 5F, G*, *Figure 5—figure supplement 1D–F*), while H3K9me3 levels at constitutive heterochromatin and on chromosome 4 were not affected (*Figure 5H, I*). Of note, while *roo* and *297* TE levels were elevated upon auxin treatment in 2.5–3 hr AEL embryos, transposon expression returned to baseline levels comparable to untreated embryos in 6–7 hr and 7–8 hr AEL time intervals (*Figure 5—figure supplement 1G*). Taken together, these data strongly indicate a relationship between the deposition of repressive H3K9me3 chromatin marks at transposon insertions and maternally deposited Piwi-piRNA complexes (*Figure 5J*).

## Discussion

Here, we have examined the role of the Piwi-directed coTGS arm of the piRNA pathway during early embryogenesis in *Drosophila*. By far, most of our insight into the function of piRNAs has derived from studies in germ cells or in the support cells of reproductive tissues. The intriguing observation that piRNAs and their Piwi-family binding partners are maternally deposited has led to speculation regarding potential roles for piRNAs in inter- and transgenerational epigenetic inheritance. Indeed, maternal piRNAs are critical in the suppression of hybrid dysgenesis induced by paternal transmission of *I*- or *P-elements* in matings with females that lack these transposons (*Brennecke et al., 2008*; *Khurana et al., 2011*). Epigenetic modifications induced by piRNAs appear to aid in piRNA cluster definition in the germline (*Akkouche et al., 2017*). Additionally, maternally deposited Aub-piRNA complexes have been implicated in embryonic gene regulation (*Barckmann et al., 2015*; *Dufourt et al., 2017*; *Rouget et al., 2010*). Last, maternally inherited piRNAs control transposon expression in interspecies hybrids between *Drosophila melanogaster* and *Drosophila* simulans (*Kelleher et al., 2012*) and regulate the TE *tirant* in the somatic compartment of the female gonad in *D. simulans* (*Akkouche et al., 2013*). Yet, to date, the lack of mechanisms to rapidly deplete maternally deposited PIWI proteins specifically from early embryos has hampered our ability to broadly assess their zygotic roles. By fusing a chemically inducible degron to Piwi, we were able to deplete Piwi-piRNA complexes from dechorionated embryos within less than 30 min of treatment and well before the nuclear accumulation of Piwi that is observed following activation of zygotic transcription.

Though nuclear localization of Piwi correlates with the appearance of its potential targets, nascent transcripts of transposons, it is unclear what triggers movement of Piwi into the somatic nuclei. Notably, nuclear translocation of Piwi lags behind in germ cell precursor nuclei, and this correlates with the observation that these nuclei activate transcription of their genomes later in embryogenesis than somatic nuclei do (*Van Doren et al., 1998*; *Zalokar, 1976*). Our dynamic imaging of Piwi localization

also revealed that it shuttles out of nuclei during mitosis, as previously observed (*Mani et al., 2014*). Since other factors of the pathway, namely components of the PICTS/SFiNX complex, remain nuclear, it is likely that Piwi is actively excluded. Several studies have previously shown that nuclear localization of Piwi is conditional upon its binding to a piRNA partners (*Klenov et al., 2011*; *Saito et al., 2009*; *Yashiro et al., 2018*), but we have no indication that Piwi is unloaded and reloaded during mitotic cycles. Rather, we hypothesize that another mechanism regulates the activity of the Piwi nuclear localization signal, though what purpose this might serve and whether it also occurs in germline and follicle cells or is restricted to embryogenesis remains unclear.

It has been suggested that the evolution of the abbreviated piRNA pathway in ovarian follicle cells arose as a consequence of the lifestyle adopted by *gypsy* family elements. These retrotransposons show their highest expression levels in the support cells surrounding the developing germline. *gypsy* family elements encode an envelope protein and have been shown to assemble into virus-like particles (*Kim et al., 1994*; *Leblanc et al., 2000*; *Song et al., 1997*). This has led to the hypothesis that their ancestral propagation strategy combined evasion of TE repression mechanisms present in germ cells with an ability to create particles that could infect the germline, where the element could insert into the genome of the developing oocyte following reverse transcription (*Kim et al., 1994*; *Leblanc et al., 2000*; *Song et al., 1997*).

While this remains speculative, it does provoke questions of whether a similar strategy is adopted by *roo* in the embryo. *roo* is a quite successful element, as indicated by it being the element with the highest copy number of individual insertions in our sequenced strains (9.4% of all identified TE insertions in the $w^{1118}$ strain and 9.9% in our degron line). How this is achieved remains mysterious since *roo* expression is extremely low in the ovary. Moreover, *roo* does not appear to be a target of the ovarian piRNA pathway since its gonadal expression is not increased nor does its HP1a enrichment and H3K9me3 levels change in piRNA pathway mutant animals (*Figure 5—figure supplement 1H, I*; *Senti et al., 2015*; *Wang and Elgin, 2011*). In the embryo, the expression of *roo* is restricted to somatic cells, especially cell lineages giving rise to the adult mesoderm. Previous studies have suggested that *roo* expression is activated by *twist* (*twi*) and *snail* (*sna*), which are highly expressed in the embryogenic mesoderm (*Brönner et al., 1995*), and this is consistent with the spatial expression pattern that we also observe. *roo* expresses the full repertoire of proteins needed to form virus-like particles, and its high expression levels (exceeding 1% of the transcriptome at its peak) might enable a strategy of propagation by infection in trans, even if rates of transmission to the germ cell precursors are relatively low.

Our data strongly suggests that only maternally deposited piRNAs engage Piwi in the soma of the developing embryo. Since *roo* is not regulated by the piRNA pathway in the ovary, evolutionary pressures must have driven the development of a set of maternal instructions that are inherited to dampen the burst of *roo* expression in the developing embryo. Indeed, 16% of maternally deposited piRNAs target *roo*. Though there are differences between the populations of piRNAs observed in ovaries as compared to embryos, this is mostly driven by the presence of follicle cell piRNAs in in samples taken from the gonad. In our small RNA analyses, we collapse all stages of oogenesis. Thus, it is not clear whether the composition of piRNA populations shifts as the ovariole matures and whether any such shifts enrich for embryonically expressed elements late in oocyte maturation. Irrespective, a set of instructions from maternal piRNA clusters clearly builds a transgenerational ability to recognize *roo* and other embryonically expressed elements.

Consistent with its recognition by Piwi-piRNA complexes and recruitment of the PICTS/SFiNX complex, H3K9me3 marks build at presumably active, euchromatic *roo* insertions as embryos progress toward stage 13 (10 hr AEL). The peak of H3K9me3 abundance lags about 2 hr behind the peak of transcription. Since we have little other information on the dynamics of piRNA-mediated silencing, it is not clear whether this is an expected observation or whether there may be mechanisms that antagonize the ability of the piRNA pathway to immediately recognize and direct heterochromatinization of expressed *roo* insertions. Of note, we see a shorter interval between the embryonic peak of *297* expression and its peak of H3K9me3 accumulation. Imposition of a repressive chromatin state on *roo* is transient during somatic development. By 13 hr AEL, H3K9me3 peaks over *roo* insertions have disappeared, but *roo* expression has not returned. The lack of H3K9me3 also correlates with the absence of critical piRNA pathway proteins in the soma. Overall, this suggests that both the expression of these TEs and the host response via small RNAs is transient. While our data provide compelling evidence of the accumulation of repressive chromatin marks at presumably actively

transcribing TE insertions, it does not carry spatial information about the precise cell types affected by H3K9me3 deposition.

Though zygotic depletion of maternal Piwi during early embryogenesis does produce a statistically significant change in *roo* expression (roughly twofold), this transposon remains highly expressed reaching up to 1% of the entire transcriptome in control animals, despite being targeted by the piRNA pathway. This provokes the question of whether targeting of *roo* by the piRNA pathway is biologically relevant. In favor of this hypothesis are several observations. *Roo* is expressed in ovaries at very low levels, yet the hallmarks of piRNA-dependent silencing, specifically H3K9me3, are absent from euchromatic *roo* insertions. This strongly indicates that *roo* is not controlled by the piRNA pathway in this tissue. Nonetheless, ovaries produce abundant *roo* piRNAs, and these are overwhelmingly in the antisense orientation. Additionally, the only uni-strand cluster expressed in germ cells, cluster 20A, has collected *roo* insertions in the antisense orientation. These piRNAs are abundantly maternally transmitted (16% of all piRNAs in embryos) and persist throughout the time during early embryogenesis when high-level *roo* expression is proposed to be driven by mesodermal transcription factors. An argument against biological significance is the lack of a clearly observable phenotype in flies following embryonic depletion of maternal Piwi. However, technical limitations enable us to only measure impacts within a single generation. It is entirely possible that the fitness cost of *roo* occupying 2% of the embryonic transcriptome might be substantial over time or in conditions flies might experience in the wild compared to the controlled rearing conditions in the lab.

Perhaps more importantly, our study demonstrates that recognition of a locus by the piRNA pathway does not necessarily impose the creation of a mitotically heritable epigenetic state. This is consistent with observations made by many groups in follicle cells wherein heterochromatin-mediated silencing of somatic transposons requires the continuous presence of the piRNA machinery (*Batki et al., 2019*; *Clark et al., 2017*; *Dönertas et al., 2013*; *Fabry et al., 2019*; *Muerdter et al., 2013*; *Murano et al., 2019*; *Ohtani et al., 2013*; *Saito et al., 2009*; *Sienski et al., 2015*; *Sienski et al., 2012*; *Zhao et al., 2019*). These data are at odds with prior observations and speculation that the maintenance of silenced epigenetic states can be primed by Piwi but maintained in a Piwi-independent mechanism throughout adult life (*Gu and Elgin, 2013*). The prior study noted these effects after only a 50% reduction in embryonic Piwi protein or RNA, using either of two different strategies. Though our induced proteolytic degradation strategy is unlikely to completely remove all Piwi protein, Piwi was reduced to levels that are undetectable by western blotting (*Figure 5B*) or via the fluorescence of its fused GFP (*Figure 5C*), which would, if anything, be expected to produce a more profound impact. While it is difficult to reconcile our observations with the interpretation of the prior study, there were substantial differences in what was measured and in how the measurements were made (i.e., a different set of genomic loci was studied in different *Drosophila* strains by different methods). The prior work made use of position-effect reporters integrated into pericentromeric heterochromatin and indicated that the expression of these in adults was sensitive to Piwi depletion in the embryo. We did not examine such a reporter, and so it remains possible that H3K9me3 marks deposited in a Piwi-dependent fashion in regions adjacent to large domains of Piwi-independent H3K9 methylation might behave differently than those deposited on active, euchromatic transposons. Considerable consistency between the two studies can be found in the prior observation that HP1a occupancy in embryos did not change substantially on several transposons studied (maximum of twofold on *HeT-A*) (*Gu and Elgin, 2013*). The prior study also failed to note large-scale changes in HP1a distribution, as a proxy for methylated H3K9, and reported only very small changes in HP1a levels on a few transposon families, as assayed in larvae by ChIP-array measurements, which collapse all insertions of a given family into a single data point. The transposons that we do identify as sensitive to Piwi during early embryogenesis do not overlap with those identified in the previous study as being mildly affected by reductions in Piwi at a later developmental stage (data not shown). This is actually consistent with our observation that the effects of profound Piwi depletion on *roo* and other TEs are transient during embryogenesis. Thus, it seems that the data themselves diverge less between the two studies than do the conclusions drawn. Of note, another recent report found a mild upregulation of transposons in pre-ZGA embryos upon maternal depletion of Piwi; however, this work relied on germ cell-specific knockdown during late stages of oogenesis rather than direct protein depletion in the embryo, thus at least some of the observed effects could stem from TE mobilization during ovary development (*Gonzalez et al., 2021*).

A recent detailed and elegant study examined the patterns of H3K9me3 accumulation during early embryogenesis in *Drosophila miranda* (*Wei et al., 2021*). Though overall, deposition of H3K9me3 did not correlate with the abundance of maternally deposited piRNAs, a set of the earliest heterochromatin nucleating elements were associated with abundant piRNAs. These targeted elements had high copy numbers and showed evidence of recent transposition activity, suggesting that they were under evolutionary pressure for robust silencing both in the ovary and the soma. It should be noted that precise nucleation sites did not necessarily overlap with abundant piRNAs, suggesting that multiple silencing mechanisms might collaborate to repress these transposon families.

Considered as a whole, our data are consistent with a role for maternally deposited piRNAs in the recognition of transposon families that have focused their expression and activity during early embryogenesis. However, our data does not support a model wherein the piRNA pathway nucleates heritable patterns of heterochromatin formation that broadly pattern the epigenetic landscape of the adult *Drosophila* soma, and this is perhaps consistent with our failure to observe consequential developmental abnormalities upon negation of embryonic Piwi function.

# Materials and methods

## Key resources table

| Reagent type (species) or resource | Designation | Source or reference | Identifiers | Additional information |
|---|---|---|---|---|
| Gene (*Drosophila melanogaster*) | *nxf2* | FlyBase | FBgn0036640 | |
| Gene (*Drosophila melanogaster*) | *panx* | FlyBase | FBgn0034617 | |
| Gene (*Drosophila melanogaster*) | *piwi* | FlyBase | FBgn0004872 | |
| Gene (*Drosophila melanogaster*) | *roo* | FlyBase | FBgn0043856 | |
| Antibody | Anti-GFP (chicken polyclonal) | Abcam | Cat# ab13970 RRID:AB_300798 | WB (1:5000) IF (1:1000) |
| Antibody | Anti-Piwi (rabbit polyclonal) | DOI: 10.1016/j.cell.2007.01.043 | N/A | WB (1:1000) IF (1:500) |
| Antibody | Anti-Tubulin (mouse monoclonal) | Abcam | Cat# ab44928, RRID:AB_2241150 | WB (1:5000) |
| Antibody | Anti-Myc tag (rabbit polyclonal) | Abcam | Cat# ab9106, RRID:AB_307014 | WB (1:1000) |
| Antibody | Anti-H3K4me2 (rabbit polyclonal) | Merck Millipore | Cat# 07-030, RRID:AB_310342 (Lot# 2971019) | ChIP (1:50) |
| Antibody | Anti-H3K9me3 (rabbit polyclonal) | Active Motif | Cat# 39161, RRID:AB_2532132 (Lot# 15617003) | ChIP (1:50) |
| Antibody | Anti-Mouse IgG Alexa Fluor 488 (goat polyclonal) | Thermo Fisher Scientific | Cat# A-11029, RRID:AB_2534088 | IF (1:500) |
| Antibody | Anti-Mouse IgG Alexa Fluor 555 (goat polyclonal) | Thermo Fisher Scientific | Cat# A-21424, RRID:AB_141780 | IF (1:500) |
| Antibody | Anti-Rabbit IgG Alexa Fluor 647 (goat polyclonal) | Thermo Fisher Scientific | Cat# A-21245, RRID:AB_2535813 | IF (1:500) |
| Commercial assay or kit | RIPA Lysis and Extraction Buffer | Thermo Fisher Scientific | Cat# 89901 | |
| Commercial assay or kit | cOmplete, Mini, EDTA-free Protease Inhibitor Cocktail | Sigma-Aldrich | Cat#11836170001 | |

*Continued on next page*

*Continued*

| Reagent type (species) or resource | Designation | Source or reference | Identifiers | Additional information |
|---|---|---|---|---|
| Commercial assay or kit | RNasin Plus RNase Inhibitor | Promega | Cat# N2615 | |
| Commercial assay or kit | Blood and Cell Culture DNA Mini Kit | Qiagen | Cat# 13323 | |
| Commercial assay or kit | RNeasy Mini Kit | Qiagen | Cat# 74106 | |
| Commercial assay or kit | NEBNext Poly(A) mRNA Magnetic Isolation Module | NEB | Cat# E7490L | |
| Commercial assay or kit | NEBNext Ultra Directional RNA Library Prep Kit for Illumina | NEB | Cat# E7420L | |
| Commercial assay or kit | NEBNext Ultra II DNA Library Prep Kit for Illumina | NEB | Cat# E7645L | |
| Commercial assay or kit | Invitrogen SuperScript IV Reverse Transcriptase | Thermo Fisher Scientific | Cat# 18090050 | |
| Commercial assay or kit | Indole-3-acetic acid sodium salt | Sigma-Aldrich | Cat# I5148-10G | |
| Commercial assay or kit | Pierce 16% formaldehyde (w/v), methanol-free | Thermo Fisher Scientific | Cat# 28908 | |
| Commercial assay or kit | MinElute PCR Purification Kit | Qiagen | Cat# 28004 | |
| Software, algorithm | Fiji | ImageJ | RRID:SCR_002285 | |
| Software, algorithm | Zeiss ZEN Imaging Software | Zeiss | RRID:SCR_018163 | |
| Software, algorithm | Proteome Discoverer 2.1 | Thermo Fisher Scientific | RRID:SCR_014477 | |
| Software, algorithm | R | RCoreTeam | N/A | |
| Software, algorithm | STAR | DOI:10.1093/bioinformatics/bts635 | RRID:SCR_015899 | |
| Software, algorithm | TEMP | https://github.com/JialiUMassWengLab/TEMP *Zhuang et al., 2014* | RRID:SCR_001788 | |
| Software, algorithm | Prodigal | https://github.com/hyattpd/Prodigal | N/A | |
| Software, algorithm | DEseq2 | DOI:10.1186/s13059-014-0550-8 | RRID:SCR_015687 | |
| Software, algorithm | Image Studio Lite | LI-COR | RRID:SCR_013715 | |

## Fly stocks and handling

All flies were kept at 25°C on standard cornmeal or propionic food. Flies expressing GFP-Nxf2 from the endogenous locus were generated by CRISPR/Cas9 (*Fabry et al., 2019*). Transgenic flies carrying a BAC transgene expressing GFP-Panx and GFP-Piwi were generated by the Brennecke lab (*Handler et al., 2013*; *Sienski et al., 2015*) and obtained from the Vienna *Drosophila* Resource Center. Control $w^{1118}$ flies were a gift from the University of Cambridge Department of Genetics Fly Facility, and flies expressing His2Av-RFP were a gift from the St Johnston lab. Flies between 3 and 14 days after hatching were used for experiments.

## Generation of fly strains

GFP-AID-Piwi knock-in flies were generated by CRISPR/Cas9 genome engineering. Homology arms of 1 kb flanking the targeting site were cloned into pUC19 by Gibson Assembly and co-injected with pCFD3 (Addgene # 49410) containing a single-guide RNA (*Port et al., 2014*) into embryos expressing vas-Cas9 (Bloomington Drosophila Stock Center # 51323). Flies expressing OsTIR1 under the *D.*

*melanogaster* Ubiquitin-63E promoter were generated by *phiC31* integrase-mediated transgenesis by injection of plasmids containing expression cassettes for proteins into embryos of genotype 'y w P[y[+t7.7]=nos-phiC31\int.NLS]X #12; +; P[y[+t7.7]=CaryP]attP2,' resulting in transgene integration on chromosome 3. Microinjection and fly stock generation was carried out by the University of Cambridge Department of Genetics Fly Facility. Transgenic and knock-in flies were identified by genotyping PCRs and confirmed via Sanger sequencing.

## Western blot

Protein concentration was measured using a Direct Detect Infrared Spectrometer (Merck). 20 µg of proteins were separated on a NuPAGE 4–12% Bis-Tris gel (Thermo Fisher Scientific). Proteins were transferred for 2 hr at 100 V, 400 mA, 100 W on an Immun-Blot Low Fluorescent PVDF Membrane (BioRad) and blocked for 1 hr in 1× LI-COR TBS Blocking Buffer (LI-COR). Primary antibodies were incubated overnight at 4°C. LI-COR secondary antibodies were incubated for 45 min at room temperature (RT) and images acquired with an Odyssey CLx scanner (LI-COR).

## *Drosophila* ovary immunofluorescence

Fly ovaries were dissected in ice-cold Phosphate-buffered saline (PBS) and fixed in 4% PFA diluted in PBS for 15 min at room temperature while rotating. Following three rinses and three 10 min washing steps in PBS-Tr (0.3% Triton X-100 in PBS), ovaries were blocked for 2 hr at RT while rotating in PBS-Tr + 1% BSA. Primary antibody incubation was carried out in blocking buffer overnight at 4°C while rotating, followed by three washing steps for 10 min each in PBS-Tr. All following steps were performed in the dark. Secondary antibodies were diluted in blocking buffer and incubated overnight at 4°C while rotating. Ovaries were washed four times for 10 min in PBS-Tr and stained with 0.5 µg/ml DAPI (Thermo Fisher Scientific) for 10 min. Following two additional washing steps for 5 min in PBS, ovaries were mounted in ProLong Diamond Antifade Mountant (Thermo Fisher Scientific) and imaged on a Leica SP8 confocal microscope using a 40× Oil objective.

## *Drosophila* embryo immunofluorescence

Embryos were collected and dechorionated in 50% bleach for 1 min. Embryos were transferred into 1 ml fixing solution (600 µl 4% PFA in PBS, 400 µl n-heptane) and fixed for 20 min at RT while rotating. The lower aqueous phase was removed and 600 µl methanol added. The tube was vortexed vigorously for 1 min to remove vitelline membranes. Embryos were allowed to sink to the bottom of the tube and all liquid was removed, followed by two washes with methanol for 1 min each. Embryos were stored at −20°C at least overnight or until further processing. In order to rehydrate embryos, three washes each 5 min with PBST (0.1% Tween20 in PBS) were performed and embryos blocked for 1 hr at RT in PBST + 5% BSA. Primary antibodies were incubated overnight at 4°C while rotating in blocking buffer followed by 3 washes for 15 min each with PBST. All following steps were performed in the dark. Secondary antibodies were diluted in blocking buffer and incubated at RT for 2 hr. Embryos were rinsed three times and washed two times for 15 min. Nuclei were stained with 0.5 µg/ml DAPI (Thermo Fisher Scientific) for 10 min. Following two additional washing steps for 5 min in PBS, embryos were mounted in ProLong Diamond Antifade Mountant (Thermo Fisher Scientific) and imaged on a Leica SP8 confocal microscope using a 40× Oil objective.

## Combined RNA-FISH and IF in embryos

Embryos were collected, dechorionated, and processed as described above until secondary antibody incubation. For all steps containing BSA addition, RNAsin Plus RNase inhibitors were added (1:1000, Promega). Following secondary antibody incubation, cells were washed three times for 15 min in PBST at RT while rotating. Embryos were fixed in 4% PFA in PBST solution for 25 min and rinsed three times with PBST for 5 min each. Embryos were pre-hybridized in 100 µl hybridization buffer (50% formamide, 5× saline-sodium citrate (SSC), 9 mM citric acid pH 6.0, 0.1% Tween20, 50 µg/ml heparin, 1× Denhardt's solution [Sigma-Aldrich], 10% dextran sulfate) for 2 hr at 65°C. Probes were hybridized in hybridization buffer supplemented with 2 nM of each FISH probe at 45°C overnight. Following washing twice with probe wash buffer (50% formamide, 5× SSC, 9 mM citric acid pH 6.0, 0.1% Tween20, 50 µg/ml heparin) for 5 min and twice for 30 min at 45°C, embryos were incubated in amplification buffer (5× SSC, 0.1% Tween20, 10% dextran sulfate) for 10 min at RT. Hairpins

were prepared as described above and embryos incubated in fresh amplification buffer with 120 nM of each probe at RT overnight in the dark. Embryos were washed twice with 5× SSC for 5 min. Nuclei were stained with 0.5 µg/ml DAPI diluted in 2× SSC for 15 min. Following washing twice with 2× SSC for 10 min, embryos were mounted in ProLong Diamond Antifade Mountant (Thermo Fisher Scientific) and imaged on a Leica SP8 confocal microscope using a 40× Oil objective.

## Light-sheet fluorescent microscopy (LSFM) of *Drosophila* embryos

Embryos were collected and dechorionated as described above. 1 ml of 1% low melting point (LMP) agarose was prepared and embryos transferred into capillaries (catalog # 100003476381, Brand) using a fitting plunger. Embryos were attempted to be positioned vertically in the capillary by twisting until agarose solidified. Capillaries were stored in PBS at RT until imaging. LSFM was performed on a Zeiss Lightsheet Z.1 (Carl Zeiss, Germany) at 25°C with a 20×/1.0 Plan-Apochromat water-immersion objective lens. Embryos were lowered carefully out of the capillary into the imaging chamber filled with PBS and positioned directly between the light-sheet illumination objectives (10×/0.2, left and right). Z-stack images for GFP and RFP (excitation at 488 and 561 nm, respectively) were acquired every 2 min for >10 hr with the lowest possible laser intensity (2.5% for GFP and 10% for RFP). Generated data was analyzed in Zeiss ZEN Imaging Software and Fiji (ImageJ).

## ChIP-seq for *Drosophila* embryos

50 µl of embryos were collected and dechorionated as described above and transferred in 1 ml Crosslinking solution (1% formaldehyde in PBS, 50% n-heptane) and vortexed on high speed for precisely 15 min. 90 µl 2.5M glycine solution was added to quench excess formaldehyde and incubated for 5 min at RT while rotating. Embryos were allowed to sink to the bottom of the tube and all liquid was removed. Embryos were washed three times for 4 min with ice-cold buffer A (60 mM KCl, 15 mM NaCl, 4 mM MgCl$_2$,15mM HEPES pH 7.6, 0.5% DTT, 1× PI) supplemented with 0.1% Triton X-100 (A-Tx buffer). All liquid was removed and embryos flash-frozen and stored at −80°C until further processing. Crosslinked embryos were transferred to a 2 ml Dounce homogenizer in 1 ml A-TBP (Buffer A + 0.5% Triton X-100). Following an additional washing step with A-TBP, embryos were lysed in 1 ml A-TBP using 10 strokes with a tight-fitting pestle. Lysate was centrifuged at 3200 g for 10 min at 4°C and supernatant removed. The pellet was resuspended in 1 ml Lysis buffer (15 mM HEPES, 140 mM NaCl, 1 mM EDTA, 0.5 mM EGTA, 1% Triton, 0.5 mM DTT, 10 mM sodium buty-rate, 0.1% sodium deoxycholate, 1× PI) and incubated at 4°C for 15 min while rotating. Following centrifugation at 3200 g for 10 min at 4°C, the pellet was washed twice with Lysis buffer and centrifuged again. All liquid was removed, and the pellet resuspended in 300 µl LB3 (10 mM Tris-HCl, pH 8, 100 mM NaCl, 1 mM EDTA, 0.5 mM EGTA, 0.1% Na-Deoxycholate, 0.5% N-lauroylsarcosine, 1× PI). Sonication was carried out using the Bioruptor pico (Diagenode) for six cycles (30 s on, 30 s off settings). Debris was removed from the chromatin-containing supernatant by spinning down at full speed for 10 min at 4°C. Prepared chromatin was either frozen down in liquid nitrogen and stored at −80°C or used immediately. 5% of the chromatin fraction was flash-frozen as an input sample. 100 µl magnetic Protein A-coupled Dynabeads (Thermo Fisher Scientific) were washed three times in 1 ml blocking solution (0.2% BSA in PBS). The blocking solution was removed using a magnetic rack. 5 µl of anti-H3K9me3 or anti-H3K4me2 polyclonal antibody was diluted in 250 µl blocking solution and incubated with 100 µl washed beads by rotating at 4°C for at least 4 hr up to overnight. The supernatant was removed and beads washed three times in blocking solution as described above. The chromatin solution was added to the beads and incubated at 4°C while rotating overnight. Following four washing steps for 2 min each using ice-cold Lysis Buffer (15 mM HEPES, 140 mM NaCl, 1 mM EDTA, 0.5 mM EGTA, 1% Triton, 0.5 mM DTT, 10 mM sodium butyrate, 0.1% sodium deoxycholate, 1× PI, 0.05% SDS), beads were washed two additional times with ice-cold 1× TE buffer. All liquid was removed and beads resuspended in 200 µl Elution buffer (50 mM Tris-HCl, pH 8; 10 mM EDTA; 1% SDS). Input samples were thawed and brought up to 200 µl with Elution buffer. Samples were transferred into 200 µl Maxymum Recovery PCR tubes (Axygen) and incubated at 65°C for 16–18 hr for reverse crosslinking. RNA contamination was removed by adding 200 µl 1× TE buffer and 8 µl of 1 mg/ml RNase A (Ambion) to ChIP and input samples followed by incubation at 37°C for 30 min. Proteins were digested using 4 µl Proteinase K (800 U/ml, NEB) and incubation at 55°C for 2 hr. Reverse crosslinked DNA was recovered using the MinElute PCR purification Kit (Qiagen) according

to the manufacturer's recommendation and eluted in 15 µl nuclease-free water. DNA recovery was verified and quantified using 1 µl for Bioanalyzer (Agilent) electrophoresis.

## ChIP-seq for *Drosophila* ovaries and heads

50 *Drosophila* ovaries were dissected in ice-cold PBS. Heads were dislodged by pouring liquid nitrogen over whole flies in a dish followed by shaking and collecting 50 µl broken-off heads in 1.5 ml tube. Samples were homogenized in 100 µl Buffer A1 (60 mM KCl, 15 mM NaCl, 4 mM MgCl$_2$,15 mM HEPES pH 7.6, 0.5% DTT, 0.5% Triton X-100, 1× PI) using a rotating pestle. The volume was brought up to 1 ml with buffer A1 and formaldehyde added to a final concentration of 1.8% for crosslinking. Samples were rotated for exactly 15 min at RT and glycine solution added to a final concentration of 225 mM. Samples were allowed to rotate for an additional 5 min and were centrifuged at 4000 g for 5 min at 4°C. The supernatant was removed, the pellet washed twice with buffer A1 and once with buffer A2 (140 mM NaCl, 15 mM HEPES pH 7.6, 1 mM EDTA, 0.5 mM EGTA, 1% Triton X-100, 0.5 mM DTT, 0.1% sodium deoxycholate, 10 mM sodium butyrate, 1× PI) at 4°C. The pellet was then resuspended in 100 µl A2 buffer supplemented with 1% SDS and 0.5% N-laurosylsarcosine and incubated at 4°C for 2 hr while shaking vigorously. Lysate was sonicated using the Bioruptor pico for 16 cycles (30 s on, 30 s off). The sonicated lysate was spun at full speed for 10 min at 4°C and the supernatant transferred to a new tube. The volume was brought up to 1 ml with A2 buffer supplemented with 0.1% SDS. Chromatin used for ChIP was precleared with 15 µl washed Protein A Dynabeads and incubated with antibody coated beads as described above. Further steps were performed as described above for embryo ChIP.

## Piwi-IP from *Drosophila* ovaries and embryos for small RNA-seq

Piwi-piRNA complexes were isolated from ovaries or from 0 to 8 hr control $w^{1118}$ embryos similar to previous reports (*Hayashi et al., 2016*; *Mohn et al., 2015*). In short, 100 µl of ovaries were dissected in PBS on ice. 100 µl of embryos were collected on grape juice agar plates and transferred to a mesh strainer. Following dechorionation in 50% bleach, embryos were washed under running tap water for at least 1 min or until bleach smell disappeared. Ovary and embryo samples were washed twice with ice-cold PBS and homogenized in 1 ml lysis buffer (10 mM HEPES pH 7.3, 150 mM NaCl, 5 mM MgCl$_2$, 10% glycerol, 1% Triton x-100, 1 mM DTT, 1 mM EDTA, 0.1 mM PMSF, 1× PI, 1:1000 RNasin [Promega]) using a 2 ml Dounce homogenizer. Material was lysed with five strokes with a loose pestle and five strokes with a tight pestle on ice. Lysate was incubated for 1 hr at 4°C while rotating and centrifuged at full speed for 10 min to pellet debris. Supernatant was transferred to a new tube and protein concentration determined by Direct Detect (Millipore). 1 mg of lysate per immunoprecipitation was used for the following steps. 50 µl Protein A Dynabeads (Thermo Fisher Scientific) were washed with lysis buffer three times for 3 min each. Washed beads were resuspended in 400 µl lysis buffer and 5 µl anti-Piwi (Hannon Lab) or rabbit IgG antibodies (Abcam, ab37415) added. Following overnight incubation at 4°C while rotating, beads were washed three times for 5 min in 500 µl lysis buffer. Antibody-coupled beads were added to lysates and volume brought up to 1 ml with lysis buffer. The solution was incubated at 4°C overnight while rotating. Supernatant was removed and saved for quality control western blotting analysis. Beads were washed six times for 10 min with 1 ml wash buffer (10 mM HEPES pH 7.3, 150 mM NaCl, MgCl$_2$, 10% glycerol, 1% Empigen BB Detergent [Merck], 1× PI). For the first wash, 1 µl RNasin was added to the wash buffer and tubes were changed between each wash. 10% of beads were set aside for quality control and 90% resuspended in 1 ml Trizol (Thermo Fisher Scientific) and stored at −80°C until further processing.

## Whole-genome sequencing

100 flies were collected in a 1.5 ml tube and frozen at −80°C for at least 1 hr. High molecular weight genomic DNA was isolated using the Blood and Cell Culture DNA Mini kit (Qiagen). Flies were homogenized using a rotating pestle on ice for 1 min. 700 µl G2 and 50 µl Proteinase K (800 U/ml, NEB) were added to each tube and incubated at 50°C for 2 hr with occasional tube inversions. Tubes were spun at 5000 g for 10 min at 4°C and supernatant transferred to new tube avoiding debris. A Qiagen Genomic-tip 20/G was equilibrated with 1 ml QBT buffer and allowed to empty by gravity flow. The supernatant containing digested proteins and genomic DNA was added to the column

and allowed to flow through. The column was washed three times with 1 ml QC buffer. Elution was carried out with 1 ml QF elution buffer and repeated once. Flow through was transferred to two new tubes (1 ml each) and 700 µl isopropanol added. Tubes were inverted 10 times and centrifuged at full speed for 15 min at 4℃. The pellet was washed with 70% ethanol twice and air-dried for 5 min. 25 µl RNase-free water was added and DNA resuspended by flicking tube gently several times while incubating at 37℃ for 2 hr. DNA was stored at 4℃. DNA was sheered using a Covaris S220 (Covaris). 3 µg of genomic DNA was diluted in RNase-Free water and transferred to a AFA Fiber Crimp-Cap (PN520052, Covaris) microtube. Sonication was carried out with the following settings: peak incident power (W) 105, duty factor 5%, cycles per burst 200, and treatment time 80 s. This resulted in sheared DNA fragments peaking at 500 bp. DNA was recovered using the QIAquick PCR Purification Kit (Qiagen).

## RNA extraction

RNA for RNA-seq and qRT-PCR experiments was isolated using the RNeasy Mini kit (Qiagen) with on-column DNA digestion (RNase-free DNase Set, Qiagen) according to the manufacturer's recommendations. RNA for small RNA-seq experiments were isolated using Trizol following the manufacturer's instructions.

## Library preparation

1 µg of total RNA was used as input material for RNA-seq library preparation. The NEBNext Poly(A) mRNA magnetic Isolation Module (NEB) was used to isolate poly(A) RNAs. Libraries were generated with the NEBNext Ultra Directional RNA Library Prep kit for Illumina (NEB) according to the manufacturer's instructions. Small RNA libraries were generated as described previously (*Jayaprakash et al., 2011*). In short, 19- to 28-nt-sized small RNAs were purified by PAGE from Piwi-bound RNA isolated from ovaries or embryos. Next, the 3' adapter (containing four random nucleotides at the 5' end) was ligated overnight using T4 RNA ligase 2, truncated KQ (NEB). Following recovery of the products by PAGE purification, the 5' adapter (containing four random nucleotides at the 3' end) was ligated to the small RNAs using T4 RNA ligase (Abcam) for 1 hr. Small RNAs containing both adapters were recovered by PAGE purification, reverse transcribed, and PCR amplified. WGS libraries were generated using the NEBNext Ultra II DNA Library Prep kit (NEB) according to the manufacturer's recommendation with 1 µg input material. Three PCR amplification cycles were performed. Libraries were quantified using the Library Quantification Kit for Illumina (Kapa Biosystems).

## Next-generation sequencing

Sequencing was performed by the Genomics Core facility at CRUK CI. RNA-seq, ChIP-seq, and small RNA-seq libraries were sequenced on an Illumina HiSeq 4000 according to the manufacturer's recommendations using single-end 50 bp runs. WGS libraries were sequenced with paired-end 150 bp runs on Illumina HiSeq 4000 or NovaSeq.

## Quantitative reverse transcription polymerase chain reaction (qRT-PCR)

Reverse transcription was performed using the SuperScript IV reverse transcriptase Kit (Thermo Fisher Scientific) with 1 µg of total RNA. qRT-PCR was performed on a QuantStudio Real-Time PCR Light Cycler (Thermo Fisher Scientific) in technical triplicates. Expression of targets was quantified using the ddCT method (*Livak and Schmittgen, 2001*). Fold-change was calculated as indicated in the figure legends and normalized to *rp49*. All primers are listed in *Supplementary file 1*.

## Protein isolation from whole embryos and quantitative mass spectrometry

100 µl of control $w^{1118}$ embryos for time points 0–2 hr, 5–7 hr, and 10–12 hr AEL were collected in three biological replicates on agar plates and dechorionated. Embryos were then lysed in lysis buffer (0.1% SDS, 0.1 M triethylammonium bicarbonate [TEAB], 1× Halt Protease and Phosphatase Inhibitor [Thermo Fisher Scientific]) using a rotating pestle on ice for 2 min or until entirely homogenized. Lysate was heated for 5 min at 90℃ and probe sonicated for 20 s (20% power with pulse of 1 s). Debris was pelleted by centrifugation at full speed for 10 min at 4℃ and supernatant transferred to

a new tube. Protein concentration was measured using Bradford Assay (BioRad). 100 μg protein was digested with trypsin overnight at 37°C. TMT chemical isobaric labeling was performed as described previously (*Papachristou et al., 2018*). Peptide fractions were analyzed on a Dionex Ultimate 3000 UHPLC system coupled with the nano-ESI Fusion Lumos mass spectrometer (Thermo Fisher Scientific).

## Treatment of embryos for auxin-induced degradation

Embryos were collected for 30 min and dechorionated. Control embryos were transferred into a fine mesh strainer placed in a plastic dish and submerged in PBS. 1 M auxin solution was generated by diluting Indole-3-acetic acid (IAA), a highly permeable small molecule as recently shown for *Caenorhabditis elegans* embryos (*Zhang et al., 2015*), in water and stored protected from light at −20°C. Auxin-treated embryos were submerged in PBS with indicated auxin concentrations. Embryos were placed at 25°C for appropriate times and harvested for RNA experiments by transferring into 1 ml Trizol followed by RNA extraction. Embryos used for ChIP-seq were processed as described above.

## RNA-seq and ChIP-seq analysis

Raw fastq files contained 50 bp reads. The first and the last two bases of all reads were trimmed using fastx_trimmer (http://hannonlab.cshl.edu/fastx_toolkit/). Reads were first aligned to the consensus sequence for all *D. melanogaster* transposons using STAR (*Dobin et al., 2013*) allowing random allocation of multimappers. Unmapped reads were further aligned to *D. melanogaster* genome release 6 (dm6) keeping uniquely mapping reads. Generated bam files for RNA-seq were further split in reads originating from sense and antisense genomic strands using samtools view options -f 0x10 and -F 0x10 for sense and antisense reads, respectively (*Li et al., 2009c*). Indexes were generated using samtools index function. Coverage files were generated using bamCoverage with normalization mode `–normalizeUsing CPM` (*Ramírez et al., 2014*) and applying a scaling factor (`–scaleFactor`). Scaling factors for individual files were calculated by dividing the sum of mapped reads contained in the file by the sum of all transposon and dm6 mapping reads of the corresponding library. Reads mapping to protein-coding genes were counted with htseq (*Anders et al., 2015*) using a feature file downloaded from Ensembl release BDGP6.22. Reads mapping to individual transposons were counted with a custom script using samtools idxstats function to extract reads mapping to individual sequences of the reference genome/transposon consensus sequence.

Count files for RNA-seq time-course experiments generated as described above were normalized to rpm to account for differences in library size and allow comparability between time points. Heatmaps displaying expression profiles of genes and transposons during embryogenesis show the mean expression values of the biological replicates, while bar graphs display the individual data points as well as the mean expression and standard deviation. Bar graphs and heatmaps were plotted in R using ggplot2.

RNA-seq experiments comparing auxin- and PBS-treated embryos of the same stage and collection were analyzed using differential expression quantification methods allowing for statistical evaluation of differences between RNA output as a direct result of auxin treatment. Differential expression analysis was performed using DESeq2 (*Love et al., 2014*). MA plots show base mean RNA expression across conditions and were calculated as previously described by Love and colleagues.

ChIP-seq reads were normalized by library size and rpm calculated for concatenated replicates using the deepTools2 bamCoverage function (*Ramírez et al., 2016*) with bin size 10 bp. MA plots displaying H3K9me3 signal intensity fold-changes between auxin-treated and control samples were calculated using DESeq2 for individual replicates (n = 3). Metaplots flanking euchromatic transposon insertion sites were calculated using computeMatrix scale-region function from deepTools2 with bin size 10 bp. All scripts used for sequencing analysis are available on GitHub (https://github.com/mhf27/hannon_roo_fabry2021, copy archived at swh:1:rev: f088572638701e0ae6f13d9e025642b9476146b5; *Fabry, 2021*).

## Small RNA-seq analysis

Reads from small RNA-seq libraries were adapter clipped using fastx_clipper with settings -Q33 -l 15 -a AGATCGGAAGAGCACACGTCT. The first and last four bases of adapter clipped reads were

trimmed using seqtk trimfq (https://github.com/lh3/seqtk; *Li, 2021*). Only high-quality reads with length between 19 and 31 bp were used for further analysis. Small RNAs were aligned as described above and size profiles plotted in R.

## Generation of annotation files

Mappability track for dm6 with 50 bp resolution was calculated according to a previously published method (*Derrien et al., 2012*). The de novo transposon insertion calling for the homozygous control $w^{1118}$ strain and our line carrying both GFP-AID-Piwi and OsTIR1 was performed using the TEMP algorithm (*Zhuang et al., 2014*). In brief, ~500 bp genomic DNA fragments were amplified and sequenced generating 150 bp paired-end reads, which were aligned to dm6 using BWA (*Li and Durbin, 2009b*). Reads with only one mate aligned to dm6 were extracted and the unmapped mate uniquely aligned to transposon consensus sequences in order to ensure correct directionality calling. Calculated insertion sites were extracted from generated GTF files if they were supported by reads on both sides (1p1). Transposon insertion files containing coordinates as well as statistical information have been submitted to GEO (GSE160778). Euchromatic regions (chr2R:6460000–25286936, chr2L:1–22160000, chr3L:1–23030000, chr3R:4200000–32079331, chrX:250000–21500000) were defined by measuring H3K9me3 signal genome-wide in sliding windows of 10 kb bins and calculating signal enrichment over input. We identified a total of 632 euchromatic TE insertions in $w^{1118}$ and 1738 in our degron strain (GFP-AID-Piwi; OsTIR1). The protein database used to identify peptides from *Drosophila* genes and transposons was generated by merging an existing database downloaded from FlyBase (dmel-r6.24.fa) with translated ORFs of transposons. ORFs were predicted and translated using prodigal (https://github.com/hyattpd/Prodigal; *Hyatt, 2020*). ORFs with less than 300 amino acids were removed using seqtk -L 300 and the file was converted to fasta format.

## Protein domain prediction

Functional analysis of protein sequences was performed using the InterPro web application (https://www.ebi.ac.uk/interpro/). Protein domains and families for ORF encoded by *roo* transcripts were predicted using default settings.

## Mass spectrometry raw data processing

Raw data files were processed according to previous reports (*Papachristou et al., 2018*). Spectral .raw files were analyzed with the SequestHT search engine on Thermo Scientific Proteome Discoverer 2.1 for peptide and protein identification. Data was searched against a modified FlyBase protein database with the following parameters: precursor mass tolerance 20 ppm, fragment mass tolerance 0.5 Da. Dynamic modifications were oxidation of methionine residues (+15.995 Da) and deamidation of asparagine and glutamine (+0.984 Da), and static modifications were TMT6plex at any amino-terminus, lysine (+229.163 Da), and methylthio at cysteine (+45.988 Da). The Reporter Ion Quantifier node included a TMT 6plex (Thermo Scientific Instruments) Quantification Method, for MS3 scan events, HCD activation type, integration window tolerance 20 ppm, and integration method Most Confident Centroid. Peptides with an FDR > 1% were removed. The downstream workflow included signal-to-noise (S/N) calculation of TMT intensities. Level of confidence for peptide identifications was estimated using the Percolator node with decoy database search. Strict FDR was set at q-value < 0.01.

## Bioinformatics analysis of proteomics data

Processed data files were analyzed as described in a previous publication (*Papachristou et al., 2018*) using qPLEXanalyzer in R with multimapping peptides included in the analysis. Bar graphs showing protein intensities for Piwi and volcano plots with indicated comparisons were plotted in R using ggplot2.

## Quantification and statistical analysis

Statistical tests used for individual experiments are indicated in the figure legends. Statistical analyses applied to hatching rates, qPCR datasets, and ChIP-seq signal intensity were calculated by unpaired (two-sample) t-test. Significance of TMT mass spectrometry data was calculated according to *Papachristou et al., 2018*. Differential expression of RNA-seq experiments and differential

enrichment of ChIP-seq experiments was calculated using DeSeq2 using adjusted p values as described in *Love et al., 2014*. The number of biological replicates is indicated in the figure legends.

## Acknowledgements

We thank Marie Bao from Life Science Editors (http://lifescienceeditors.com) for comments on the manuscript. We thank the members of the Hannon group, especially Susanne Bornelöv, for discussion and feedback on the manuscript. We thank the Cancer Research UK Cambridge Institute Bioinformatics, Genomics, Microscopy, RICS, and Proteomics Core Facilities for support, in particular Kamal Kishore, Evangelia Papachristou, and Carmen Gonzalez Tejedo. We thank the University of Cambridge Department of Genetics Fly Facility for microinjection services and fly stock generation. We thank the Vienna *Drosophila* Resource Center for fly stocks. We thank Daniel St Johnston for H2Av-RFP flies. This research was funded in whole, or in part, by Cancer Research UK (A21143) and the Wellcome Trust (110161/Z/15/Z). For the purpose of open access, the author has applied a CC BY public copyright license to any Author Accepted Manuscript version arising from this submission. GJH is a Royal Society Wolfson Research Professor (RP130039 and RSRP\R\200001).

## Additional information

### Funding

| Funder | Grant reference number | Author |
|---|---|---|
| Royal Society | RP130039 | Gregory J Hannon |
| Royal Society | RSRP\R\200001 | Gregory J Hannon |
| Cancer Research UK | A21143 | Gregory J Hannon |
| Wellcome Trust | 110161/Z/15/Z | Gregory J Hannon |

The funders had no role in study design, data collection and interpretation, or the decision to submit the work for publication.

### Author contributions

Martin H Fabry, Conceptualization, Data curation, Software, Formal analysis, Supervision, Validation, Investigation, Visualization, Methodology, Writing - original draft, Writing - review and editing; Federica A Falconio, Validation, Investigation, Writing - review and editing; Fadwa Joud, Data curation, Formal analysis, Investigation, Visualization, Writing - review and editing; Emily K Lythgoe, Investigation, Writing - review and editing; Benjamin Czech, Conceptualization, Data curation, Supervision, Validation, Investigation, Visualization, Methodology, Writing - original draft, Project administration, Writing - review and editing; Gregory J Hannon, Conceptualization, Supervision, Funding acquisition, Methodology, Writing - original draft, Project administration, Writing - review and editing

### Author ORCIDs

Martin H Fabry ⓘ https://orcid.org/0000-0002-8484-4715
Benjamin Czech ⓘ https://orcid.org/0000-0001-8471-0007
Gregory J Hannon ⓘ https://orcid.org/0000-0003-4021-3898

### Decision letter and Author response

Decision letter https://doi.org/10.7554/eLife.68573.sa1
Author response https://doi.org/10.7554/eLife.68573.sa2

## Additional files

### Supplementary files

• Supplementary file 1. List of oligonucleotides for qRT-PCR and RNA-FISH.

- Transparent reporting form

## Data availability

Raw data from high-throughput sequencing experiments are available at GEO under accession number GSE160778. Raw data from proteomics experiments are available on PRIDE with accession number PXD022409. Source data files have been provided for Figures 1 and 2.

The following datasets were generated:

| Author(s) | Year | Dataset title | Dataset URL | Database and Identifier |
|---|---|---|---|---|
| Fabry MH, Falconio FA, Joud F, Lythgoe EK, Czech B, Hannon GJ | 2021 | Maternally inherited piRNAs direct transient heterochromatin formation at active transposons during early *Drosophila* embryogenesis | https://www.ncbi.nlm.nih.gov/geo/query/acc.cgi?acc=GSE160778 | NCBI Gene Expression Omnibus, GSE160778 |
| Fabry MH, Falconio FA, Joud F, Lythgoe EK, Czech B, Hannon GJ | 2021 | Maternally inherited piRNAs silence transposons during *Drosophila* embryogenesis | https://www.ebi.ac.uk/pride/archive/projects/PXD022409 | PRIDE, PXD022409 |

The following previously published dataset was used:

| Author(s) | Year | Dataset title | Dataset URL | Database and Identifier |
|---|---|---|---|---|
| Fabry MH, Ciabrelli F, Munafò M, Eastwood EL, Kneuss E, Falciatori I, Falconio FA, Hannon GJ, Czech B | 2019 | piRNA-guided co-transcriptional silencing coopts nuclear export factors | https://www.ncbi.nlm.nih.gov/geo/query/acc.cgi?acc=GSE121661 | NCBI Gene Expression Omnibus, GSE121661 |

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
