## [Decision Letter]

**Acceptance summary:**

This manuscript is of broad interest to readers in the fields of RNA biology, epigenetics, and early development. The authors provide an exceptionally comprehensive description of the temporal dynamics of mobile element RNA, host defense protein, and epigenetic mark abundance across fruit fly early embryo development. Pairing this descriptive work with the first application of a rapid protein degradation system to maternal proteins in the model fruit fly embryo, the authors reject a previously accepted model that the maternally-deposited protein Piwi establishes gene silencing transmitted epigenetically to later stages of development.

**Decision letter after peer review:**

Thank you for submitting your article "Maternally inherited piRNAs silence transposons during *Drosophila* embryogenesis" for consideration by *eLife*. Your article has been reviewed by 3 peer reviewers, and the evaluation has been overseen by a Reviewing Editor and Kevin Struhl as the Senior Editor. The reviewers have opted to remain anonymous.

The reviewers have discussed their reviews with one another, and the Reviewing Editor has drafted this to help you prepare a revised submission. The three reviewers applauded this exceptionally comprehensive description of the temporal dynamics of transposable element RNA, of piRNA pathway proteins and of H3K9me3 across embryogenesis. The reviewers also agreed that the novel application of the auxin-degron system to maternally deposited proteins in the early embryo opens new avenues of research for the community. Finally, the inference that maternal Piwi may actually not set up epigenetic silencing in later life stages represents an important course correction for the field. You will see below, however, that the reviewers were unconvinced that the current data support the major claim of the paper articulated in title, abstract, and final model, namely, that maternal Piwi is required for transposon suppression. We welcome a revised version of the manuscript that stays closer to the data, guided by the list of essential revisions below.

Essential revisions:

New experiments/analyses

1) The reviewers were unconvinced that the data support the major claim of the paper- namely, that maternally-deposited Piwi silences roo. In wildtype embryos, the dramatic increase in roo by 4hr and especially the sharp increase between 2hr-4hr – compared to the comparatively subtle 2-fold increase upon Piwi degradation – makes careful developmental staging hugely important for inferences from the latter experimental data. An internal control to carefully calibrate the RNA-seq with progression of development is necessary. Moreover, the 297 transposon curiously shows just over 2-fold increase versus control. How can we reconcile this change with log10 mean expression is >3 on the Figure 5D but in Figure 1C, the 297 transposon RPMs are barely registering any RPMs throughout embryogenesis.

2) Related to point 1), the reviewers requested additional timepoints for the auxin-induced degradation experiment. Specifically, RNA-seq or even just qPCR well-after the timepoints reported for both roo and 297. These timeports are important for determining if these two elements still drop to original low levels by 12hr or 17hr despite Piwi degradation at the beginning of embryogenesis. If so, the major claim of the paper would be yet further undermined.

3) The reviewers appreciated the compelling loss of H3K9me3 across the genome and at roo insertions upon Piwi degradation. However, the reviewers were less convinced of the significance of H3K9me3 depletion for gene regulation. For example, in Figure 5-SuppFig-1D, the roo insertion with H3K9me3 is overlapping Hid promoter, it seems surprising that there is little change in Hid mRNA levels after auxin induced degradation and loss of H3K9me3 signal. Related, for Figure 3B, the authors should overlay their RNAseq data with the ChIP-seq tracks for lbm and Tsp42El (which are both normally expressed in the embryo). If the pattern is like what is shown on Flybase, lbm and Tsp42El expression may actually increase during embryogenesis in the same degree of H3K9me3 accumulation around the roo insertion. This pattern could oppose the model of direct or meaningful silencing by piRNAs. While there is clearly a piRNA-chromatin response at roo insertions, the effect on roo silencing may actually be quite modest, and the modesty of this effect may contribute to the perplexing lack of any later developmental phenotypes from the auxin-induced Piwi degradation.

4) The reviewers agreed that one of the most impactful contributions made by this report is the rejection of the model put forward by Gu and Elgin. To further probe what may account for the differences between the two studies, the authors could take advantage of their own data. Specifically, in Gu and Elgin, they found different TE families responded differently to the depletion of maternal Piwi – authors could pull these TE families from their genome-wide data to investigate the dynamics (expression and K9 enrichment in response to maternal piwi depletion), further addressing this discrepancy.

Language modification/softening claims/key clarifications

5) Even if the 2-fold increase in roo holds up after more rigorously controlled developmental staging, the reviewers remained unconvinced that such a subtle change (compared to the dramatic spike in roo expression WT embryos) warrants the current title (and the model in figure 5). Unless, for example, the 2-fold increase triggers additional roo transposition (as assayed by WGS), then the title/model appears overstated. Modification of this claim in the title, abstract, and discussion is required.

6) In the ovary, roo is the transposon family with the highest density of antisense piRNAs present and roo mRNA is strongly upregulated upon combined Aub/Ago3 knock-down (Senti et al., 2015). By contrast the authors state that roo is not regulated by the ovarian piRNA pathway (lines 760-763). Their statement, however, is based only on nxf2 KO flies, which inhibit specifically coTGS and not PTGS, in line with previous findings of roo being insensitive to Piwi-mediated regulation in the ovary (Théron, NAR 2018). A revision of this interpretation and referencing the papers showing potent piRNA-mediated regulation of ovarian roo transcripts by PTGS is necessary.

7) The authors' finding of 297 is interesting but needs more elaboration. Based on previous functional studies of piRNA coTGS mutants (e.g., piwi, mael, arx, mael), 297's response is categorized with TE families that are classified as "opposite categories" by authors' data and interpretation – 412 and mdg1 families. In these previous studies, similar to 412 and mdg1, 297 has burst transcription and reduced K9 in these mutants in ovarian somatic cells. I am puzzled by how to reconcile the authors' interpretation of Roo and 297 with these previous findings. Based on Figure1 – SfigF, the expression dynamics of 297 differs from that of roo. Can this be attributed to the absence of normalization by copy number?

8) Since roo is known to be potently regulated through PTGS in the ovary and Aub is maternally inherited, conclusions about somatic piRNA-mediated regulation of roo (and other PTGS-targeted transposons) requires depleting Aub in addition to Piwi. If this experiment is not feasible, a description of the limitations of Piwi depletion specifically regarding likely redundancy between embryonic coTGS and PTGS by Piwi- and Aub-piRNA complexes, respectively, is essential.

9) Images requires quantification. In Figure 2-supplement 1, for example, by only glancing at the images for piwi, panx, and nxf2 vs H2Av localization, I would not have drawn the same conclusion as the authors. Authors should quantify the co-localization of GFP and RFP foci to support their conclusion.

10) The reviewers found omissions in the methods section that require attention.

a. Given the challenges of diffusing small molecules across dechorionated embryos, additional detail about the development of the auxin systems is warranted.

b. Transposon-calling method in the *w1118* genotype and the strain with the AID-tagged Piwi should be reported. In the auxin-degron study, the sequenced strain would be heterozygous, complicating TE calling with short reads. Might the uncertainty associated with calling TEs in heterozygotes have led to the confusing results in Figure 5D and E? (For Figure 5 – SFigure C-E, the H3K9me3 enrichment is not right at the boundary of 297 insertions, but a short distance from it). This is a sharp contrast to Figure 3A-B, which is more typical -are the 297 boundaries inappropriately shifted?

c. It appears that only euchromatic TEs were incorporated into the analysis – if so, please clearly state this.

d. The y-axes should be the same for Figure 2A (for piwi) and Figure 2-SFigE (for panx) and 2-SFigG (for Nxf2) to help with comparison across these factors. Same for Figure 2B and Figure 2-SF and H.

e. Details of the ChIP-seq analyses are missing. For some, the authors used rpm (e.g., Figure 3A) while at other places, authors used fold enrichment (e.g., Figure 5E-was the former not normalized to input while the latter was?)

11) What prompts the massive clearance of the H3K9me across the 177 roo insertions after 10h AEL and does this have a real link to Piwi/piRNA binding to the roo nascent transcripts? Maybe speculate on this more in the discussion?

12) Pg 9 Line 236 To determine whether roo might be competent for retrotransposition in embryos, the authors mined quantitative proteomic data for roo peptides of gag, pol and env, but this just establishes ORF expression, not the act of retrotransposition, which actually requires WGS analysis for new copies of roo TE insertions. I suggest changing to more accurate statement like "To determine whether roo mRNAs are effectively being translated during the pulse of embryonic expression, we mined quantitative proteomic data…"

13) What is the consequence of lower viability of dechorionated embryos in regard to RNA-seq and ChIP-seq analyses. When do the embryos die off and how would this affect the dynamic range of the analyses?

14) Calling the orientation of TEs from short read data is tricky. How were these data validated?

*Reviewer #1:*

This study provides the most solid characterization of *Drosophila* Piwi mRNA and protein levels throughout embryogenesis to date, with insightful data on *Drosophila* embryonic TE expression and H3K9m3 marks, and useful creation of a GFP-AID-Piwi fly strain that enables testing whether maternally contributed Piwi has a direct role in responding to TE expression and H3K9me3 levels in the embryos after auxin-induced Piwi degradation. I applaud the thorough analysis, well written prose, beautiful figures and movies, and well-designed experiments. All the data and reagents from this study need to be shared with the public in a revision of the manuscript that I would welcome to see.

But I have one major contention with the hard push of authors to fit the data to a desired hypothesis and mechanism which proposes that maternally-deposited Piwi/piRNAs have a direct role in "silencing" the roo transposon. This issue begins with the paper's title of "Maternally inherited piRNAs silence transposons during *Drosophila* embryogenesis". My critiques and interpretation of the data suggest to me that Maternal Piwi and piRNAs are 'Responding' to the major roo transposon expression burst during *Drosophila* embryogenesis, but the importance of a silencing role is debatable.*Reviewer #2:*

In this study, Fabry and co-authors combined elegant and novel genetic experiments, live imaging, and functional genomics to investigate the role of maternally deposit piRNA machinery, in particular piwi, in silencing transposable elements (TEs) in developing embryos. Surprisingly, they found that maternally deposited piwi, but not zygotically expressed piwi, plays a dominant role in silencing TE families that are predominantly active in the embryos, especially for Roo and 297. Their developmental time course analysis provides a detailed investigation for the sequential events during embryogenesis, namely the nuclear localization of piwi, the burst of TE transcription, and transcriptional silencing of TEs. While the authors' findings are exciting and seem well supported, there are some incongruencies with previous studies, and these need to be further addressed. This includes that piwi's role in maintaining, in addition to initiating, TE epigenetic silencing in somatic tissues and whether TE family 297's activities and host-directed silencing are predominantly embryonic. The varying dynamics among different TE families may worth further investigation (by performing analyses that are normalized by TE copy number) to gain a full picture of the role of piwi in suppressing not only Roo, but also other TE families. Some technical details also need further clarification. Overall, this is an important study that will further our understanding of how hosts suppress selfish genetic parasites.

*Reviewer #3:*

Argonaute proteins of the Piwi-clade are best known for their role in germline silencing of transposons, but both Piwi and Aub have been shown to also be highly expressed in somatic cells during early *Drosophila* embryogenesis raising the question of putative somatic regulatory functions. Fabry et al. address this important question by first performing a detailed characterization of endogenous expression of transposons and the transposon silencing piRNA pathway during *Drosophila* embryogenesis (Figures 1-3) and then an experimental test of the embryonic functions of maternally inherited Piwi/piRNA molecules using a new protocol for degron-mediated protein depletion in dechorionated embryos (Figures 4-5).

In the first part of the paper the authors describe strong embryonic expression of the roo transposon as well as embryonic expression of maternally inherited Piwi. Although this to a certain extent validates previous observations, the authors' comprehensive analyses of transposons and piRNA pathway genes in both transcriptome and proteome data adds a very valuable and novel overview of this gene regulation. In addition, the authors extend on the current knowledge in several places, for example by characterizing Piwi expression from maternal vs zygotic origin (Figure 2E-F).

In the second part of the paper the authors present an elegant implementation of auxin-mediated depletion of degron-tagged Piwi. By auxin administration to dechorionated embryos, the authors are able to deplete maternally inherited Piwi-piRNA protein complexes within 25 minutes. This is an important advance compared to previous RNAi-based methods. In the following analyses the authors describe Piwi-dependent embryonic heterochromatin formation at roo and 297 transposon insertions.

The experiments are in general well designed and controlled and the analyses are broad and comprehensive. As the authors highlight in the last paragraph of the paper, the presented data largely disproves the tested hypothesis of Piwi-piRNA-mediated somatic epigenetic gene regulation in the *Drosophila* embryo. This finding is important and will be of broad interest for the field.

My main reservation with the current manuscript is that I find the conclusions on embryonic transposon regulation to be unnecessarily overstated. I find that this overstatement somewhat overshadows the important findings that resolve the question of maternal Piwi-piRNA functions in embryo gene regulation.

[Editors' note: further revisions were suggested prior to acceptance, as described below.]

Thank you for submitting your article "Maternally inherited piRNAs direct transient heterochromatin formation at active transposons during early *Drosophila* embryogenesis" for consideration by eLife.

We found the revised language much more in line with the data and were satisfied with virtually all additional analyses, particularly the new data on embryo staging. A few hanging concerns remain that I trust can be quickly addressed. Note that the absence of tracked changes in the revision document made it difficult, at least in two instances, to track stated adjustments to the text. Please point to these changes with line numbers.

Essential Revisions:

1. The referees requested that panel C from the figure for reviewers be included in the main text along with A and B (maybe as part of the supplement to figure 5? (rebuttal point 1)).

2. Please include language justifying the use of different approaches to analyzing the RNA-seq data presented in Figures 1C and 5D (rebuttal point 1).

3. Please add to the main text, possibly in the legend, that the different strains (*w1118* and the degron strain) had different 297 insertion numbers/mean expression (rebuttal point 1).

4. Please point to where language referring to "further highlighted the different normalization strategies in…the text" (rebuttal point 1) is found.

5. Please point to where language referring to "Appropriate adjustments have been made to the text underlining the limitations of whole embryo approaches…" is found.

6. Given that the only major publication addressing maternal Piwi impacts on epigenetic silencing uncovered a very different result, additional language in the main text reconciling the current dataset with Gu and Elgin is still warranted. The few sentences added to the revision are not sufficient to help the reader understand the discrepancy. The cited more modest depletion of Piwi in Gu and Elgin should have more modest effects on the Piwi-regulated TEs- the absence of overlap with your more complete Piwi depletion remains to be explained. More, the discovery of no overlap between upregulated TEs in the two datasets lacks reference to the new figure (or at least a parenthetical "data not shown").

---

## [Author Response]

Essential revisions:-New experiments/analyses1) The reviewers were unconvinced that the data support the major claim of the paper- namely, that maternally-deposited Piwi silences roo. In wildtype embryos, the dramatic increase in roo by 4hr and especially the sharp increase between 2hr-4hr – compared to the comparatively subtle 2-fold increase upon Piwi degradation – makes careful developmental staging hugely important for inferences from the latter experimental data. An internal control to carefully calibrate the RNA-seq with progression of development is necessary.

First, regarding the main claim of the paper, perhaps we used language that was a bit too strong, and we appreciate the feedback from the referees in that regard. We have now changed the title and calibrated that language throughout the paper. However, we don’t believe that it is appropriate to abandon entirely the notion that *roo* is regulated by maternally deposited piRNAs in *Drosophila* embryos. *Roo* expression is essentially restricted to embryogenesis, yet, flies produce abundant *roo* piRNAs, including from a specialized piRNA cluster (20A), which is the only uni-strand cluster expressed in the germline. This cluster has gathered *roo* insertions in predominantly the antisense orientation, as has *flamenco* for *gypsy* family elements. The evolutionary pressure to create and maintain such a locus strongly suggests a biologically relevant regulatory relationship between the piRNA pathway and *roo*. *Roo* piRNAs are the most abundant class deposited in embryos, where H3K9me3 marks are indisputably deposited on *roo* insertions in a Piwi-dependent fashion. *Roo* does show a strong spike in expression in early embryos, which is increased further in the absence of Piwi. Thus, the piRNA pathway does blunt or dampen the expression of *roo* during this developmental stage. Though loss of that regulation does not have an overt phenotypic impact in a single generation in optimal lab conditions, it remains a strong possibility that the selective pressure to retain recognition of *roo* by the piRNA pathway in embryos implies that there could be fitness costs should that link be broken. Therefore, while we have modified the language of the paper and attempted to make the points raised herein clearer, we continue to believe that our data does indicate piRNA-dependent regulation of *roo*.

Regarding the possibility of technical artifacts that might arise from variations in the developmental stages of the embryos at our various time points, we have taken great precautions to limit artifacts arising from female egg retention by ensuring optimal egg laying conditions as well as manually checking a portion of sampled embryos for their expected stage. However, in order to further validate our staging strategy and as requested by the referees, we downloaded publicly available data sets from Flybase profiling gene expression during the complete embryogenesis of *Drosophila* (Graveley et al., 2011) and compared gene expression with our sampled time points in control *w1118* embryos. We found strong correlation in the expression of genes between our experiments (included as an image for the reviewers but subsequently added as Figure 1—figure supplement 1A) and the reference data (included as an image for the reviewers but subsequently added as Figure 1—figure supplement 1B). By examining well-studied genes for primarily maternally deposited transcripts such as the maternal *alphaTub67C* or *exuperantia* (*exu*), involved in proper localisation of *bicoid* mRNA during oogenesis, and early zygotic transcripts such as the mesodermal master transcription factor *twist* (*twi*), which has been shown to drive *roo* expression, the regulator of dorsal-ventral pattern *zerknullt* (*zen*) and *disrupted underground network* (*dunk*), involved in syncytial embryo cellularisation during stage 5 (2-3h AEL), we further validated our early time points spanning the beginning of embryogenesis and zygotic genome activation (ZGA). Of note, we found very similar expression pattern between our 2.5-3h AEL *w1118* embryo timepoint compared to the degradation experiment profiling 2.5-3h AEL GFP-AID-Piwi; OsTIR1 embryos with or without auxin treatment (included as an image for the reviewers but subsequently added as Figure 5—figure supplement 1A).

Moreover, the 297 transposon curiously shows just over 2-fold increase versus control. How can we reconcile this change with log10 mean expression is >3 on the Figure 5D but in Figure 1C, the 297 transposon RPMs are barely registering any RPMs throughout embryogenesis.

Figure 1C illustrates the expression of transposon transcripts for all sampled embryo time points in our control *w1118* strain. Our data is normalised to reads per million (rpm). Figure 5D shows the base mean expression (in log_10_) of the experiment on the x-axis and fold-change on the y-axis. The base mean is calculated using a different normalisation strategy compared to rpm (median of ratios: counts divided by sample-specific size factors determined by median ratio of gene counts relative to geometric mean per gene) as described in (Love et al., 2014). Therefore, rpm and log_10_ base mean are not directly comparable. Additionally, our *w1118* control strain used for Figure 1C has a lower number of *297* insertions than the degron strain used for Figure 5D (27 euchromatic insertions in *w1118* versus 82 insertions in the GFP-AID-Piwi; OsTIR1 line). The mean expression of *297* in *w1118* is ~70 rpm while mean expression in untreated degron embryos is 167 rpm (converted from log_10_ base means). This discrepancy is likely due to the different transposon content in both strains. In order to avoid confusion of the reader, we further highlighted the different normalisation strategies in both the text and method sections.

2) Related to point 1), the reviewers requested additional timepoints for the auxin-induced degradation experiment. Specifically, RNA-seq or even just qPCR well-after the timepoints reported for both roo and 297. These timeports are important for determining if these two elements still drop to original low levels by 12hr or 17hr despite Piwi degradation at the beginning of embryogenesis. If so, the major claim of the paper would be yet further undermined.

We believe that we have addressed the reviewer’s concerns by performing qPCR experiments on embryos derived from our degron strain (GFP-AID-Piwi; OsTIR1) treated with or without auxin for additional time points as requested in order to assess whether *roo* and *297* expression levels remain elevated over control samples or return to baseline expression. By analysing expression of both *roo* and *297* from early embryogenesis to later stages, we were able to replicate our previous finding showing upregulation of both *roo* and *297* upon auxin treatment in 2.5-3h embryos. However, both TE expression levels return to baseline values at 6-7h and 7-8h AEL time points (thus we did not perform the experiment at time points as late as 12h or 17h AEL). These data suggest that transposon expression upon Piwi depletion is only affected in early embryogenesis. We included this data in our revised manuscript as Figure 5—figure supplement 1F and discuss the results in context of the overall conclusions drawn from our new data. We do disagree that this further undermines the major claim of the paper. This simply means that *roo* and *297* are either silenced by other means at those developmental time points or that the transcription factors necessary to drive their expression during a specific window of development are simply absent. While the transcription factors regulating *297* are not known, for *roo* the latter possibility is supported by the expression of *twist* (*twi*), which shows strongest expression between 2-6h AEL (also see Figure 1—figure supplement A,B). Note, some delay in *roo* expression is expected due to the dynamics of translation and activation of *roo* loci by Twi protein.

3) The reviewers appreciated the compelling loss of H3K9me3 across the genome and at roo insertions upon Piwi degradation. However, the reviewers were less convinced of the significance of H3K9me3 depletion for gene regulation. For example, in Figure 5-SuppFig-1D, the roo insertion with H3K9me3 is overlapping Hid promoter, it seems surprising that there is little change in Hid mRNA levels after auxin induced degradation and loss of H3K9me3 signal. Related, for Figure 3B, the authors should overlay their RNAseq data with the ChIP-seq tracks for lbm and Tsp42El (which are both normally expressed in the embryo). If the pattern is like what is shown on Flybase, lbm and Tsp42El expression may actually increase during embryogenesis in the same degree of H3K9me3 accumulation around the roo insertion. This pattern could oppose the model of direct or meaningful silencing by piRNAs. While there is clearly a piRNA-chromatin response at roo insertions, the effect on roo silencing may actually be quite modest, and the modesty of this effect may contribute to the perplexing lack of any later developmental phenotypes from the auxin-induced Piwi degradation.

The reviewers are correct in noting that Hid expression seems not affected by degradation of Piwi protein and subsequent loss of H3K9me3 marks during ZGA (Figure 5—figure supplement 1D). However, our experiments were performed on whole embryos. As we showed in our FISH experiments (Figure 1—figure supplement 1B), *roo* is primarily expressed in specific tissues associated with the developing mesoderm. H3K9me3 deposition as a response to Piwi-mediated silencing is therefore only likely to occur in tissues actually showing *roo* expression, as H3K9me3 is thought to occur co-transcriptionally. Therefore, expression of genes in proximity of *roo* insertions are only silenced if they are co-expressed in cells together with *roo,* which it does not appear to be. Our ChIP-seq and RNA-seq data show an average of H3K9me3 occupancy and gene expression respectively of the whole embryo. These data illustrate the advantages of “clean” systems like OSCs (cell culture) in which loss of the piRNA pathway has a uniform impact on gene expression upon loss of H3K9me3.

Gene expression of *lbm* and *Tsp42EI* followed patterns observed in previous studies (Graveley et al., 2011), which showed onset of transcription in later stages of embryogenesis (8-9h AEL). However, H3K9me3 signal accumulated at a strain-specific *roo* insertion in line with the overall dynamics observed in our study (Author response image 1). While accumulation of H3K9me3 was detected in our experiments, this data lacks spatial information and H3K9me3 accumulating cells might not overlap cells expressing both examined genes. These technical challenges will be overcome in the future by the development of single-cell chromatin profiling methods as well as multiomics approaches. Appropriate adjustments have been made to the text underlining the limitations of whole embryo approaches for both profiling RNA and chromatin modifications in bulk embryos.

**Author response image 1. sa2fig1:** Gene expression and H3K9me3 occupancy of genes in proximity of euchromatic TE insertions. IGV genome browser screenshot showing H3K9me3 ChIP-seq signal for the indicated genes on chromosome 2R carrying a *w1118*-specific *roo* insertion. Blue tracks show gene expression for indicated embryo time points (in rpm; n=2). Green tracks illustrate H3K9me3 occupancy at comparable time points (in rpm; n=2).

4) The reviewers agreed that one of the most impactful contributions made by this report is the rejection of the model put forward by Gu and Elgin. To further probe what may account for the differences between the two studies, the authors could take advantage of their own data. Specifically, in Gu and Elgin, they found different TE families responded differently to the depletion of maternal Piwi – authors could pull these TE families from their genome-wide data to investigate the dynamics (expression and K9 enrichment in response to maternal piwi depletion), further addressing this discrepancy.

The study by Gu and Elgin investigates the impact of Piwi reduction in early embryos on further developmental stages beyond embryogenesis using *piwi* null mutants. Piwi depletion in early embryos was achieved by crossing heterozygous null mutants (*piwi^2^*/+ x *piwi^2^*/+) therefore reducing, but not abolishing, maternally deposited Piwi protein (Figure 2A, Gu and Elgin, 2013). The authors analysed the impact of Piwi reduction in embryogenesis by evaluating HP1a binding and H3K9me3 levels in larvae using ChIP-array for selected genomic targets including TEs in two replicates (Figure 5B and S6, Gu and Elgin, 2013). Some transposons did not show strong HP1a enrichment in wild-type larvae (including *roo*, *DMRP1* and *XDMR*), while other TEs such as *gypsy5*, *invader3* and *DIVER2* showed a higher enrichment over the input (> 2-fold). Most TEs showed an overall modest decrease in HP1a levels (69 out of 83 classes) including highlighted classes such as *Bari1*, *Invader1*, *mdg1*, *Het*-*A* and *TART* in larvae that inherited reduced Piwi levels. H3K9me3 levels at *roo* sequences showed no enrichment over input in wild type larvae (Figure S6, Gu and Elgin, 2013) in accordance with our observation that H3K9me3 is lost at later stages of development (Figure 3, this study). While Gu and Elgin assess the impact of maternal Piwi reduction during embryogenesis in a later larval stage, our study focuses on illuminating the impact of maternally deposited Piwi-piRNA complex on early embryogenesis. However, in order to compare both studies, we highlighted TE classes reported in the study by Gu and Elgin (using the same colour scheme used in Gu and Elgin) in our data evaluating expression profiles and H3K9me3 occupancy in treated versus untreated embryos during ZGA (Author response image 2), as requested by the reviewers. TE classes reported by Gu and Elgin as sensitive to Piwi depletion in larvae did not show significant changes in either RNA expression or H3K9me3 occupancy in our study. These results are likely due to differences in our depletion strategy targeting the entire pool of maternally deposited proteins rather than reducing Piwi levels by only 2-fold as well as the different read out (RNA-seq and ChIP-seq in our study versus HP1a ChIP-array in Gu and Elgin’s study). Additionally, different developmental stages were evaluated in both studies (early embryogenesis, this study versus larvae, Gu and Elgin) further complicating comparisons. Last, our analyses are based different transposon annotations with the data reported here using annotations containing the full length TE sequences, while the study by Gu and Elgin used separate versions for transposon bodies and LTRs annotations.

**Author response image 2. sa2fig2:** Transposons sensitive to Piwi reduction in larvae showed no change in our study. (A) MA plot showing base mean expression (log_10_ scale) of transposon RNAs relative to their fold-change (log_2_ scale) in GFP-AID-Piwi; OsTIR1 embryos treated with 5mM auxin versus control (n=3). Blue=TE classes with low HP1a enrichment in wild type larvae as reported in Gu et al., 2013. Brown=TE classes with reported HP1a enrichment >2 fold over input in wild type larvae as reported in Gu et al., 2013. Red=TE classes with reported loss of HP1a enrichment in *piwi^2^/piwi^2^* mutants vs control as reported in Gu et al., 2013. (B) MA plot showing base mean signal intensity (log_10_ scale) of TEs relative to the H3K9me3 ChIP-seq signal enrichment (log_2_ scale) in GFP-AID-Piwi; OsTIR1 embryos treated with 5mM auxin versus control (n=3). Colour scheme same as in (A).

Language modification/softening claims/key clarifications5) Even if the 2-fold increase in roo holds up after more rigorously controlled developmental staging, the reviewers remained unconvinced that such a subtle change (compared to the dramatic spike in roo expression WT embryos) warrants the current title (and the model in figure 5). Unless, for example, the 2-fold increase triggers additional roo transposition (as assayed by WGS), then the title/model appears overstated. Modification of this claim in the title, abstract, and discussion is required.

We acknowledge the reviewer’s concern about overstating the silencing capacity of a maternally inherited piRNA-dependent mechanism on transposon expression in early embryogenesis. However, our data clearly shows statistically significant upregulation of *roo* and *297* as well as loss of H3K9me3 during ZGA of embryogenesis upon auxin-mediated Piwi depletion. Whether this is biologically relevant remains to be determined (though see above), which we address in our discussion. Both *297* and *roo* steady-state RNA levels returned to baseline at later time points. This indicates that while the piRNA pathway engages transposons and apparently dampens transcription initially during embryogenesis, it fails to contain the surge of transcription following ZGA likely in response to increased abundance of transcription factors such as twist and snail further driving expression. We included our new findings (see point 2) in the revised manuscript and discuss the implications in greater detail. Additionally, we adjusted our claims/model accordingly.

We also included a paragraph in the discussion which explicitly raises whether *roo* regulation by piRNAs is biologically relevant, so that readers can weight the evidence for themselves:

“Though zygotic depletion of maternal Piwi during early embryogenesis does produce a statistically significant change in *roo* expression (roughly 2-fold), this transposon remains highly expressed reaching up to 1% of the entire transcriptome in control animals, despite being targeted by the piRNA pathway. This provokes the question of whether targeting of *roo* by the piRNA pathway is biologically relevant. In favour of this hypothesis are several observations. *Roo* is expressed in ovaries at very low levels, yet the hallmarks of piRNA-dependent silencing, specifically H3K9me3, are absent from euchromatic *roo* insertions. This strongly indicates that *roo* is not controlled by the piRNA pathway in this tissue. Nonetheless, ovaries produce abundant *roo* piRNAs, and these are overwhelmingly in the antisense orientation. Additionally, the only uni-strand cluster expressed in germ cells, cluster 20A, has collected *roo* insertions in the antisense orientation. These piRNAs are abundantly maternally transmitted (16% of all piRNAs in embryos) and persist throughout the time during early embryogenesis when high level *roo* expression is proposed to be driven by mesodermal transcription factors. An argument against biological significance is the lack of a clearly observable phenotype in flies following embryonic depletion of maternal Piwi. However, technical limitations enable us to only measure impacts within a single generation. It is entirely possible that the fitness cost of *roo* occupying 2% of the embryonic transcriptome might be substantial over time or in conditions flies might experience in the wild compared to the controlled rearing conditions in the lab.”

6) In the ovary, roo is the transposon family with the highest density of antisense piRNAs present and roo mRNA is strongly upregulated upon combined Aub/Ago3 knock-down (Senti et al., 2015). By contrast the authors state that roo is not regulated by the ovarian piRNA pathway (lines 760-763). Their statement, however, is based only on nxf2 KO flies, which inhibit specifically coTGS and not PTGS, in line with previous findings of roo being insensitive to Piwi-mediated regulation in the ovary (Théron, NAR 2018). A revision of this interpretation and referencing the papers showing potent piRNA-mediated regulation of ovarian roo transcripts by PTGS is necessary.

Indeed, Senti et al., 2015 report that a high proportion of antisense Piwi-bound piRNAs are mapping to *roo* in line with our study (Figure 2G, this study) as well as previous reports (Brennecke et al., 2009). Knockdown of *aub* and *ago3* led indeed to a strong decrease in *roo*-mapping Piwi-bound piRNAs (Figure 2C, Senti et al.). The authors additionally show that most *roo*-mapping piRNAs (>90%) are antisense to the TE’s consensus sequence (Figure 2D, Senti et al.) thereby identifying the origin of these piRNAs as likely cluster-derived rather than a product of PTGS.

However, we were unable to find the basis of the reviewer’s statement that *roo* mRNA is strongly upregulated upon double knockdown of *aub* and *ago3*. The authors report steady-state mRNA levels for all knockdown conditions. Mining supplementary table S1, we were able to compare expression levels for *roo* in germline knockdowns (GLKD) of *white*, *piwi*, *aub*, *ago3* and *aub/ago3* since *roo* expression levels are not reported or discussed in the manuscript. *roo* showed a base expression (*white* GLKD) of 0.411 RPKM. This increased to 0.572 in *piwi* GLKD, 0.788 in *aub* GLKD, 0.864 in *ago3* GLKD and 1.644 in *aub/ago3* GLKD (Supplementary Table S1, Senti et al.). While the fold-change between *aub/ago3* versus *white* GLKD is precisely 4, we argue that this is likely irrelevant due to the extremely low baseline expression in *white* GLKD, which is just over the detection limit. These data are in complete agreement with our reports of low expression of *roo* in heterozygous *nxf2* mutants (0.04 RPKM) versus homozygous mutants (0.06 RPKM). We included Senti et al., 2015 in our citations as further evidence that *roo* is neither affected by loss of piRNA-dependent coTGS (*nxf2* mutants, Figure 5—figure supplement 1G, this study; *piwi* GLKD Senti et al., 2015) nor by loss of PTGS (*aub*, *ago3*, *aub/ago3* GLKD, Senti et al., 2015). We apologise if our conclusions based on revisiting data in Senti et al. are incorrect and would ask the referees to highlight relevant information should we have missed it.

7) The authors' finding of 297 is interesting but needs more elaboration. Based on previous functional studies of piRNA coTGS mutants (e.g., piwi, mael, arx, mael), 297's response is categorized with TE families that are classified as "opposite categories" by authors' data and interpretation – 412 and mdg1 families. In these previous studies, similar to 412 and mdg1, 297 has burst transcription and reduced K9 in these mutants in ovarian somatic cells. I am puzzled by how to reconcile the authors' interpretation of Roo and 297 with these previous findings. Based on Figure1 – SfigF, the expression dynamics of 297 differs from that of roo. Can this be attributed to the absence of normalization by copy number?

It is correct that *297* has been previously reported to be controlled by the piRNA pathway in *Drosophila* OSCs (Sienski et al., 2012). Loss of coTGS factors such as *nxf2* or *panx* leads to strong transcriptional de-repression accompanied by loss of H3K9me3 at TE insertions (Sienski et al., 2015, Batki et al., 2019, Murano et al., 2019, Fabry et al., 2019, Zhao et al., 2019). Our data shows a very similar behaviour during embryogenesis. In Figure 3—figure supplement 1B we show the accumulation of H3K9me3 at euchromatic *297* insertions during embryogenesis as well as for ovaries and heads. Indeed, the chromatin state shows different dynamics compared to *roo* H3K9me3 profiles (Figure 3A). Unlike *roo*, which is not regulated by the piRNA pathway in ovaries (see point 6), *297* acquires a typical H3K9me3 signature at TE insertions in ovaries, likely as a result of coTGS in somatic sheet cells, which are the parent tissue of OSCs. These findings are consistent with the current model of coTGS in which germline regulated TEs are silenced and H3K9me3 levels increase around affected insertions. *297* showed a lower number of *297* insertions in our control *w1118* strain (n=20) compared to *roo* insertions (n=117). Discrepancies in chromatin dynamics between the two TE classes are likely a result of different transcriptional control mechanisms. While the driver of *297* transcription is unknown, *roo* expression is likely to be controlled by the mesoderm-associated transcription factors twist and snail (Bronner et al., 1995), which show strong expression during early stages of embryogenesis and match the *roo* expression profile.

8) Since roo is known to be potently regulated through PTGS in the ovary and Aub is maternally inherited, conclusions about somatic piRNA-mediated regulation of roo (and other PTGS-targeted transposons) requires depleting Aub in addition to Piwi. If this experiment is not feasible, a description of the limitations of Piwi depletion specifically regarding likely redundancy between embryonic coTGS and PTGS by Piwi- and Aub-piRNA complexes, respectively, is essential.

Our analysis has indicated that *roo* is not under the control of either piRNA-dependent coTGS or PTGS in ovaries (see point 6 – but we await further information from the referee). However, Aub depletion together with Piwi is not yet feasible. Aub and Piwi are located on the same chromosome within a distance of 10kb thereby making recombination of CRISPR-mediated AID-Aub and AID-Piwi alleles highly unlikely. Engineering the Aub locus in the AID-Piwi fly could overcome these limitations, however it is not an established strategy and beyond the scope of this study.

We agree that maternally deposited Aub-piRNA complexes might serve a significant function during embryogenesis especially in regards of germline development since maternally deposited Aub primarily localises to germ plasm during embryogenesis and is not enriched in somatic nuclei (Brennecke et al., 2009). However, for these complexes to regulate embryonic transposons by PTGS, they would likely have to do so by an RNAi-like cleavage mechanism, since Ago3 is not strongly maternally deposited and we do not detect strong ping-pong signatures in embryonic piRNA populations. Of course, such signals could be swamped if Aub only performs this function in pole cells.

9) Images requires quantification. In Figure 2-supplement 1, for example, by only glancing at the images for piwi, panx, and nxf2 vs H2Av localization, I would not have drawn the same conclusion as the authors. Authors should quantify the co-localization of GFP and RFP foci to support their conclusion.

We analysed the co-localisation of Piwi, Panx, and Nxf2 with H2Av and found striking co-localisation of the three studied GFP-tagged coTGS factors and RFP-tagged H2Av as displayed by measurements of fluorescence intensity across sections for both channels in Figure 2—figure supplement 1C-E.

10) The reviewers found omissions in the methods section that require attention.a. Given the challenges of diffusing small molecules across dechorionated embryos, additional detail about the development of the auxin systems is warranted.

The auxin-mediated degradation system has been intensively characterised in multiple animal models including *C. elegans*, *Drosophila* and mammalian systems. Auxin is a small molecule and multiple studies suggest high cellular permeability. Additionally, a recent study found high permeability of auxin through the egg and cuticle, which are well known to show poor drug efficacy due to limited permeability, even during embryogenesis (Zhang et al., 2015). We added this information as well as the appropriate citation in the methods section of the revised manuscript.

b. Transposon-calling method in the w1118 genotype and the strain with the AID-tagged Piwi should be reported. In the auxin-degron study, the sequenced strain would be heterozygous, complicating TE calling with short reads. Might the uncertainty associated with calling TEs in heterozygotes have led to the confusing results in Figure 5D and E? (For Figure 5 – SFigure C-E, the H3K9me3 enrichment is not right at the boundary of 297 insertions, but a short distance from it). This is a sharp contrast to Figure 3A-B, which is more typical -are the 297 boundaries inappropriately shifted?

Transposons were called as described in our methods section using the TEMP algorithm. We include all relevant information and a comprehensive description of our strategy in line 1141-1147 of our original manuscript. More detailed information about the strategy can be found in the TEMP publication (Zhuang et al., 2014), which we cite in the methods section. We included only high confidence transposon insertions that were supported by at least one unique read at both sides (reference genome and TE consensus sequence) in order to ensure the information on orientation is correct. To the best of our knowledge TE insertions have been accurately called using our very deeply sequenced WGS data. Files containing transposon insertions including relevant statistical measurements have been submitted as supplementary files to GEO (accession number GSE160778). We sequenced only homozygous flies of our control *w1118* and degron strain (GFP-AID-Piwi; OsTIR1) and all our experiments have been performed using homozygous embryos or flies.

c. It appears that only euchromatic TEs were incorporated into the analysis – if so, please clearly state this.

Yes, this is correct. We would like to point out that our original manuscript clearly stated the use of only euchromatic transposons in our analyses in the text, figure legends, and methods section (see line 436, 457, 677, 1129 of the original manuscript). We have nevertheless highlighted this at other occasions that did not contain this information earlier.

d. The y-axes should be the same for Figure 2A (for piwi) and Figure 2-SFigE (for panx) and 2-SFigG (for Nxf2) to help with comparison across these factors. Same for Figure 2B and Figure 2-SF and H.

While we do not agree with this comment, we have updated the figures for RNA-seq and MS data using the same y-axes as requested. We would like to stress that the purpose of these panels were not to compare the expression of the highlighted genes among each other but across the studied time course.

e. Details of the ChIP-seq analyses are missing. For some, the authors used rpm (e.g., Figure 3A) while at other places, authors used fold enrichment (e.g., Figure 5E-was the former not normalized to input while the latter was?)

ChIP-seq experiments reported in Figure 3 and Figure 3—figure supplement 1 show H3K9me3 signal in reads per million (rpm) calculated for 10bp genome wide bin sizes using the deepTools2 bamCoverage function (Ramírez et al., 2016). Figure 5E reports fold-changes between H3K9me3 levels of auxin versus control treated embryos. Differential chromatin state analysis was performed using DESeq2 as reported in Love et al., 2014. We highlighted the appropriate information more clearly in the methods section and figure legends.

11) What prompts the massive clearance of the H3K9me across the 177 roo insertions after 10h AEL and does this have a real link to Piwi/piRNA binding to the roo nascent transcripts? Maybe speculate on this more in the discussion?

We added the following to the section in which this result is presented:

“This is consistent both with the known requirement for active transcription for targeting by Piwi and with the observed need for continuous engagement of PICTS to maintain H3K9me3 marks on transposon loci.”

12) Pg 9 Line 236 To determine whether roo might be competent for retrotransposition in embryos, the authors mined quantitative proteomic data for roo peptides of gag, pol and env, but this just establishes ORF expression, not the act of retrotransposition, which actually requires WGS analysis for new copies of roo TE insertions. I suggest changing to more accurate statement like "To determine whether roo mRNAs are effectively being translated during the pulse of embryonic expression, we mined quantitative proteomic data…"

We changed the wording accordingly.

13) What is the consequence of lower viability of dechorionated embryos in regard to RNA-seq and ChIP-seq analyses. When do the embryos die off and how would this affect the dynamic range of the analyses?

Dechorionation using bleach exposes the embryo surface leading to increased sensitivity to dehydration and leakage through mechanical stress. We did not observe any biases of embryos failing to develop due to dechorionation in regard to stages of development.

14) Calling the orientation of TEs from short read data is tricky. How were these data validated?

Transposon calling was performed on paired-end 150bp data. The mean fragment size of our DNA fragments used for library preparation were >500bp to allow for sequencing of fragments overlapping the reference sequence as well as the unique internal sequence of our TE consensus sequences. By allowing mapping of only unique pairs, we excluded the possibility of including fragments spanning the reference sequence and only the LTR sequence found at the beginning and end of the TE in our analysis thereby ensuring high confidence in orientation calling.

[Editors' note: further revisions were suggested prior to acceptance, as described below.]

We found the revised language much more in line with the data and were satisfied with virtually all additional analyses, particularly the new data on embryo staging. A few hanging concerns remain that I trust can be quickly addressed. Note that the absence of tracked changes in the revision document made it difficult, at least in two instances, to track stated adjustments to the text. Please point to these changes with line numbers.Essential Revisions:1. The referees requested that panel C from the image for the reviewers be included in the main text along with A and B (maybe as part of the supplement to figure 5? (rebuttal point 1)).

Panel C of image for the reviewers has now been added to Figure 5—figure supplement 1 as panel A and we refer to this figure in the Results section on page 22 (lines 643-647).

2. Please include language justifying the use of different approaches to analyzing the RNA-seq data presented in Figures 1C and 5D (rebuttal point 1).

Figure 1C illustrates the expression of transposon transcripts for all sampled embryo time points in our control *w1118* strain. To consider different library sizes and facilitate comparability throughout our time course experiment (for which we only had two biological replicates per time point), we normalised our RNA-seq data to reads per million (rpm). We now state this on page 5, lines 177-179.

Considering the experiment using embryo collections either treated with auxin or PBS as a control was carried out with three biological replicates, we chose a different normalisation strategy to calculate differential expressed genes and transposons (Figure 5D). This allowed for use of the most commonly accepted current statistical modelling (DESeq2) of differential gene expression measurements of the impact of auxin treatment on RNA output (see also comment 4 below). We have modified the text in the main manuscript on page 22 (lines 640-643) to clearly state that we carried out differential expression analysis for the Piwi degradation experiments.

3. Please add to the main text, possibly in the legend, that the different strains (w1118 and the degron strain) had different 297 insertion numbers/mean expression (rebuttal point 1).

For clarity, we have included the number of degron strains- and *w1118*-specific transposon insertions, respectively, in the main text and relevant figure legends:

– Main text, page 15, lines 469-470,

– Legend of figure 3, page 16, lines 490 and 492,

– Legend of figure 3—figure supplement 1, page 17, lines 501 and 503 and 504,

– Main text, page 18, lines 531-533,

– Main text, page 22, lines 673-677,

– Legend of figure 5, page 23, line 702,

– Legend of figure 5—figure supplement 1, page 25, lines 722 and 726 and 727.

4. Please point to where language referring to "further highlighted the different normalization strategies in…the text" (rebuttal point 1) is found.

In addition to the adjustments we made in the main text of the revised manuscript (see comment 2), we also updated the methods section (lines 1233-1245) to clearly state the different normalisation strategies we used for the different datasets. The methods section now states:

“Count files for RNA-seq time course experiments generated as described above were normalised to reads per million (rpm) to account for differences in library size and allow comparability between time points. Heatmaps displaying expression profiles of genes and transposons during embryogenesis show the mean expression values of the biological replicates, while bar graphs display the individual data points as well as the mean expression and standard deviation. Bar graphs and heatmaps were plotted in R using ggplot2.

RNA-seq experiments comparing auxin- and PBS-treated embryos of the same stage and collection were analysed using differential expression quantification methods allowing for statistical evaluation of differences between RNA output as a direct result of auxin treatment. Differential expression analysis was performed using DESeq2 (Love et al., 2014). MA plots show base mean RNA expression across conditions and were calculated as previously described by Love and colleagues.”

5. Please point to where language referring to "Appropriate adjustments have been made to the text underlining the limitations of whole embryo approaches…" is found.

We have included a paragraph discussing the limitation of whole embryo approaches in the discussion on pages 27-28 (lines 837-840) of the revised manuscript:

“While our data provide compelling evidence of the accumulation of repressive chromatin marks at presumably actively transcribing TE insertions, it does not carry spatial information about the precise cell types affected by H3K9me3 deposition.”

6. Given that the only major publication addressing maternal Piwi impacts on epigenetic silencing uncovered a very different result, additional language in the main text reconciling the current dataset with Gu and Elgin is still warranted. The few sentences added to the revision are not sufficient to help the reader understand the discrepancy. The cited more modest depletion of Piwi in Gu and Elgin should have more modest effects on the Piwi-regulated TEs- the absence of overlap with your more complete Piwi depletion remains to be explained. More, the discovery of no overlap between upregulated TEs in the two datasets lacks reference to the new figure (or at least a parenthetical "data not shown").

We added text to elaborate on the discrepancy between our work and the study by Gu and Elgin in the discussion on pages 28-29 (lines 863-904). The paragraph reads:

“Perhaps more importantly, our study demonstrates that recognition of a locus by the piRNA pathway does not necessarily impose the creation of a mitotically heritable epigenetic state. […]Thus, it seems that the data themselves diverge less between the two studies than do the conclusions drawn. Of note, another recent report found a mild upregulation of transposons in pre-ZGA embryos upon maternal depletion of Piwi, however, this work relied on germ cell-specific knockdown during late stages of oogenesis rather than direct protein depletion in the embryo, thus at least some of the observed effects could stem from TE mobilisation during ovary development (Gonzalez et al., 2021).”